# Improving Block-Wise LLM Quantization by 4-bit Block-Wise Optimal Float (BOF4): Analysis and Variations

**Patrick Blumenberg, Thomas Graave & Tim Fingscheidt**
Institute for Communications Technology
Technische Universität Braunschweig
Braunschweig, 38106, Germany
{patrick.blumenberg,thomas.graave,t.fingscheidt}@tu-bs.de

## Abstract

Large language models (LLMs) demand extensive memory capacity during both fine-tuning and inference. To enable memory-efficient fine-tuning, existing methods apply block-wise quantization techniques, such as NF4 and AF4, to the network weights. We show that these quantization techniques incur suboptimal quantization errors. Therefore, as a first novelty, we propose an optimization approach for block-wise quantization. Using this method, we design a family of quantizers named 4-bit *block-wise optimal float* (BOF4), which consistently reduces the quantization error compared to both baseline methods. We provide both a theoretical and a data-driven solution for the optimization process and prove their practical equivalence. Secondly, we propose a modification to the employed normalization method based on the *signed* absolute block maximum (BOF4-S), enabling further reduction of the quantization error and empirically achieving less degradation in language modeling performance. Thirdly, we explore additional variations of block-wise quantization methods applied to LLMs through an experimental study on the importance of accurately representing zero and large-magnitude weights on the one hand, and optimization towards various error metrics on the other hand. Lastly, we introduce a mixed-precision quantization strategy dubbed *outlier-preserving quantization* (OPQ) to address the distributional mismatch induced by outlier weights in block-wise quantization. By storing outlier weights in 16-bit precision (OPQ) while applying BOF4-S, we achieve top performance among 4-bit block-wise quantization techniques w.r.t. perplexity.

## 1 Introduction

Driven by scaling the transformer architecture to billions of parameters, large language models (LLMs) have achieved remarkable performance in language modeling tasks. However, their size poses significant challenges for deployment, particularly in memory-constrained settings. Numerous post-training quantization (PTQ) techniques have been developed to mitigate this, reducing the memory footprint of the weights and, in many cases, improving inference speed (Frantar et al., 2023; Lin et al., 2024; Xiao et al., 2023; Liu et al., 2024). Fine-tuning imposes even greater memory demands, making it challenging to adapt LLMs on consumer-grade GPU hardware. To address this, Dettmers et al. (2023) introduced QLoRA, a memory-efficient fine-tuning method that combines 4-bit quantization of pre-trained weights with low-rank adaptation (LoRA) (Hu et al., 2022). For quantization, Dettmers et al. (2023) propose 4-bit NormalFloat (NF4), a quantization method with a fixed codebook. This method normalizes blocks of network weights by their absolute maximum (block-wise absmax normalization). Unlike calibration-data-based PTQ methods, NF4 is data-free. This means that NF4 quantizes weights without computing network activations, making the quantization process itself far more efficient in terms of both time and memory, while maintaining acceptable accuracy degradation at 4 bits per weight. Dettmers et al. (2023) claim that the NF4 codebook is information-theoretically optimal due to its equal utilization of the 16 reconstruction levels. However, Yoshida (2023) demonstrates that this claim is incorrect. We add that equal

utilization of reconstruction levels is not a theoretically justified criterion for the optimality of a quantizer. Yoshida (2023) also proposes an alternative codebook (AF4) designed to address the shortcomings of NF4.

In this work, we show that neither NF4 nor AF4 minimizes the quantization error of the network weights. For the first time, we provide a rigorous mathematical analysis of block-wise absmax quantization and explore multiple design variations through an experimental study. As a first contribution, we derive an expectation-maximization (EM) algorithm inspired by Lloyd's algorithm (Lloyd, 1982) that computes the correct, information-theoretically optimal codebook for block-wise absmax quantization w.r.t. the mean absolute error (MAE) or mean squared error (MSE) criterion. Additionally, we propose an alternative normalization technique: Instead of normalizing blocks by their absolute maximum value, we normalize by the signed absolute maximum. This simple modification results in a significant reduction of the quantization error. Using our EM algorithm, we compute a family of optimal codebooks which we refer to as 4-bit block-wise optimal float (BOF4), or BOF4-S when signed normalization is used. Furthermore, we identify that block-wise absmax quantization is sensitive to outlier weights affecting the distribution of the normalized weights. We address this by introducing an outlier-preserving quantization (OPQ) that stores outliers in 16-bit precision. When combined with BOF4-S, OPQ substantially improves perplexity over NF4 and AF4.

The paper is structured as follows: Section 2 reviews related work on block-wise quantization. Section 3 outlines our mathematical analysis and novel quantization methods. Section 4 details the experimental setup, and Section 5 presents and discusses the results. We conclude in Section 6.

## 2 RELATED WORK

Block-wise quantization based on blocks of input values normalized by their absolute maximum was introduced by Dettmers et al. (2022) as a method for quantizing optimizer states during neural network training. Subsequent works (Dettmers et al., 2023; Yoshida, 2023; Dotzel et al., 2024) applied this technique to LLM network weights for memory-efficient fine-tuning. We refer to this quantization method as *block-wise absmax quantization*.

### 2.1 BLOCK-WISE ABSMAX QUANTIZATION

In *block-wise absmax quantization* network weights $w_{b,i} \in \mathbb{R}$ are first grouped into blocks, with block indices $b \in \mathcal{B} = \{1, \ldots, B\}$, and indices of weights within a block $i \in \mathcal{I} = \{1, \ldots, I\}$, where $B \in \mathbb{N}$ is the number of blocks, and $I \in \mathbb{N}$ the block size. Then, the weights are normalized by the absolute maximum weight in their respective block:

$$w_b^{\max} = \max_{i \in \mathcal{I}} |w_{b,i}|, \quad b \in \mathcal{B} \tag{1}$$

$$x_{b,i} = \frac{w_{b,i}}{w_b^{\max}} \in [-1, 1], \quad i \in \mathcal{I}, b \in \mathcal{B} \tag{2}$$

Next, each normalized weight $x_{b,i}$ is quantized independently using scalar quantization. The absolute block maxima $w_b^{\max}$, commonly referred to as *quantization constants*, are stored in addition to the quantized weights for later decoding. Overall, the block-*dependent* quantization function $Q_b()$ for weights $w_{b,i}$ is defined as

$$Q_b(w_{b,i}) = w_b^{\max} \cdot \tilde{Q}(\frac{w_{b,i}}{w_b^{\max}}) = w_b^{\max} \cdot \tilde{Q}(x_{b,i}), \tag{3}$$

where $\tilde{Q}()$ is a block-*independent* quantization function.

### 2.2 4-BIT BLOCK-WISE QUANTIZATION FOR LLMS

In this section, we discuss the previous block-wise absmax quantization that our work builds upon.

**4-bit NormalFloat (NF4)** : NF4 (Dettmers et al., 2023) is a 4-bit scalar quanitzer for block-wise absmax quantization. The $L = 2^4 = 16$ reconstruction levels $\hat{x}(\ell)$, $\ell \in \mathcal{L} = \{1, \ldots, L\}$ are computed based on quantiles of the assumed Gaussian network weight distribution $p_W = \mathcal{N}(0, \sigma^2)$.

Dettmers et al. (2023) claim that their construction leads to equal utilization of the 16 reconstruction levels. However, this was already shown to be incorrect by Yoshida (2023). Furthermore, an equal probability for all codebook points is not a general criterion for the optimality of a quantizer. Instead, quantization aims at rate-distortion optimality (Berger, 2003). Accordingly, a codebook assigning equal probability to each codebook point is only optimal for uniformly distributed input data. This has been well-known for decades, most prominently through the necessary conditions for optimality that underpin Lloyd's algorithm (Lloyd, 1982).

**4-Bit AbnormalFloat (AF4):** Yoshida (2023) analyzes the distribution of normalized network weights and performs direct minimization of the mean absolute error (MAE) to obtain a block-wise absmax quantization codebook for normally distributed network weights, named AF4. This quantizer aims to correct an oversight in the design of NF4 (Dettmers et al., 2023), which does not account for the dependence of the distribution of normalized weights on the block size. However, Yoshida's optimization method targets the minimum MAE of *normalized weights* $\text{MAE}(x_{b,i}, \tilde{Q}(x_{b,i}))$, instead of minimizing the end-to-end quantization error of the network weights $\text{MAE}(w_{b,i}, Q_b(w_{b,i}))$.

Both NF4 and AF4 contain reconstruction levels at -1, 0, and 1, such that the weight of the largest absolute value in a block is represented in full 16-bit precision, while the zero is represented without error. Not including these reconstruction levels leads to significantly worse MAE, mean squared error (MSE), and perplexity. We confirm this in Appendix B.

## 3 METHODS

In this section, we introduce our methods for optimizing 4-bit block-wise absmax quantization.

### 3.1 NOVEL BLOCK-WISE *Signed* ABSMAX NORMALIZATION

Instead of normalization by the absolute block maximum, as described in Section 2.1 and commonly used in existing quantization methods such as NF4 and AF4, we propose a different normalization approach: *block-wise signed absmax normalization*. In NF4 and AF4, two reconstruction levels are intentionally constrained to $\hat{x}(1) = -1$ and $\hat{x}(16) = 1$, respectively, ensuring that the weight $w_b^{(\text{max})}$ with the largest magnitude in each block $b$ is quantized without error. This empirically improves quantization performance. However, we notice that for network weights in general position, each block $b$ of the normalized weights $x_{b,i}$ contains only one of the two endpoints, either $-1$ or $1$. This observation motivates the idea to normalize the weights by the *signed* absolute block maximum. By doing so, normalization always maps $w_b^{(\text{max})}$ to 1, instead of to 1 or $-1$ with equal probability. This preserves exact quantization of $w_b^{(\text{max})}$ while constraining only $\hat{x}(16) = 1$, as illustrated in Fig. 1. The additional free reconstruction level $\hat{x}(1)$ adds a degree of freedom for optimizing the non-uniform codebook, thereby improving the achievable quantization error. Formally, in block-wise signed absmax normalization, the quantization constants $w_b^{\text{max}}$ from (1) are selected by

$$w_b^{\text{max}} = w_{b,j^*} \quad \text{with} \quad j^* = \arg\max_{i \in \mathcal{I}} |w_{b,i}|, \quad b \in \mathcal{B}. \tag{4}$$

Except for this modification, we proceed with quantization as before, using (2) and (3). Note that no runtime overhead is incurred during inference or fine-tuning as decoding of the quantized weights is unchanged.

### 3.2 NOVEL 4-BIT BLOCK-WISE OPTIMAL FLOAT (BOF4 / BOF4-S)

To determine optimal quantization codebooks w.r.t. the MSE and MAE criteria, we design an expectation-maximization (EM) algorithm based on Lloyd's algorithm (Lloyd, 1982), a well-known algorithm for quantizer design. In each maximization step, the reconstruction levels $\hat{x}(\ell) \in \mathbb{R}$, $\ell \in \mathcal{L} = \{1, \dots L\}$ are set to the centroids of their respective Voronoi region $\mathcal{R}_\ell = [\xi(\ell-1), \xi(\ell))$, with decision boundaries $\xi(\ell)$, $\ell \in \mathcal{L}^{(\xi)} = \{0, 1, \dots, L\}$, where $\xi(0) = -\infty$ and $\xi(L) = \infty$. However, in block-wise absmax quantization, the codebook is applied to normalized weights $x_{b,i}$, whereas our goal is to minimize the quantization error of the quantized unnormalized weights $Q_b(w_{b,i})$ relative

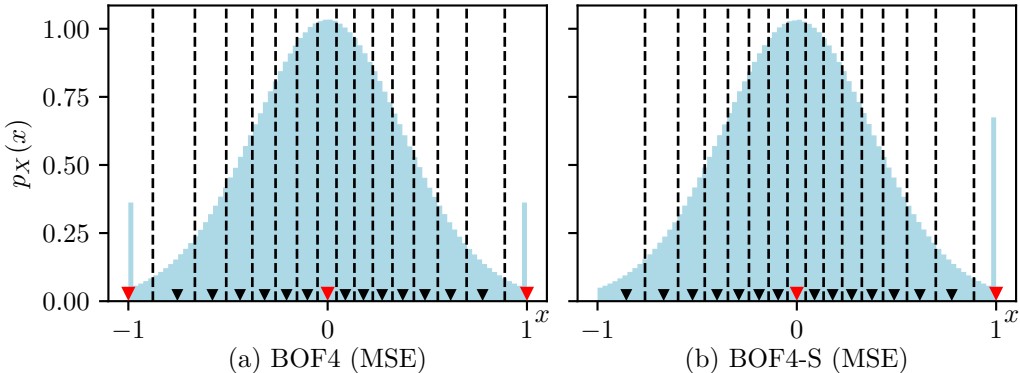

Figure 1: The blue histograms show the **distributions of normalized weights** $p_X(x)$ for block-wise *absolute* absmax normalization (left) and block-wise *signed* absmax normalization (right) assuming Gaussian network weights. Also shown are the **resulting reconstruction levels** $\hat{x}(\ell)$ (▼ fixed, ▼ optimized) and decision thresholds $\xi(\ell)$ (dashed lines), after minimizing the MSE$(\mathbf{W}, \mathbf{Q}(\mathbf{W}))$ for normally distributed network weights $\mathbf{W} = (w_{b,i})$ with $w_{b,i} \sim p_W = \mathcal{N}(0, 1)$ and block size $I = 64$. For absolute absmax normalization, we compute the 4-bit block-wise optimal float (**BOF4**, left), requiring three fixed reconstruction levels (-1, 0, 1). In contrast, when using *signed* normalization, we obtain **BOF4-S** (right), in which the largest absolute value in a block and zero are precisely represented by only two fixed reconstruction levels (0, 1), which reduces the quantization error.

to the original weights[1] $w_{b,i}$. This introduces a mismatch between the optimization target and the weight distribution directly used in Lloyd's algorithm. To resolve this, we mathematically derive an optimal solution for the centroid update. We name the resulting quantizer 4-bit block-wise optimal float (BOF4). When *signed* absmax normalization is used, we refer to it as BOF4-S. A complete derivation is provided in Appendix D, resulting codebooks in Appendix E, major results follow here.

**MSE**: Let $W$ be a random variable representing a continuous, zero-symmetric distribution of network weights. We further define two derived random variables $X$ and $M$ representing the normalized weights and absolute block maxima, respectively. Our goal is to find a reconstruction level $\hat{x}(\ell)$ that minimizes the MSE quantization error for those network weights that fall into a fixed region $\mathcal{R}_\ell$ after block-wise absmax normalization. By analytical optimization (Appendix D.2.1, (26)), we obtain the solution for the updated centroid as

$$\hat{x}(\ell) = \frac{\int_0^\infty m^2 \cdot \mathbb{E}_X[X \mid M\!=\!m, X \in \mathcal{R}_\ell] \cdot p_M(m) \cdot \left[F_X(x \mid M\!=\!m)\right]_{\xi(\ell-1)}^{\xi(\ell)} \mathrm{d}m}{\int_0^\infty m^2 \cdot p_M(m) \cdot \left[F_X(x \mid M\!=\!m)\right]_{\xi(\ell-1)}^{\xi(\ell)} \mathrm{d}m}, \quad (5)$$

where the probability density function (PDF) $p_X$ of the normalized weights, the cumulative distribution function (CDF) $F_X$ of normalized weights, and the expectation $\mathbb{E}[X \mid M\!=\!m, X \in \mathcal{R}_\ell]$ can be computed directly from the known CDF $F_W$ and PDF $p_W$ of the network weights, see Appendix D.2.1 (31). A detailed derivation and simplified solution for the special case of Gaussian network weights is also provided in Appendix D.2.1 (see (34)). Equation (5) can be solved by numerical integration. Alternatively, the centroid can be approximated by Monte-Carlo estimation based on samples drawn from the distribution $p_W$ of network weights as (see Appendix D.3 (64))

$$\hat{x}(\ell) = \frac{\sum_{k \in \mathcal{K}_\ell} w_k^2 \cdot x_k}{\sum_{k \in \mathcal{K}_\ell} w_k^2}, \quad (6)$$

where $x_k \in \mathcal{R}_\ell$ are the normalized weights that fall into region $\mathcal{R}_\ell$, $k \in \mathcal{K}_\ell = \{1, \ldots, K_\ell\}$ being their indices, and $w_k$ is the absolute block maximum $w_b^{\max}$ of the block $b$ containing $x_k$.

---

[1]Minimizing the quantization error of normalized weights $x_{b,i}$ leads to worse perplexity; see Appendix F.

**MAE**: A similar optimization can be performed for the MAE criterion, as detailed in Appendix D.2.2 (59), yielding

$$\int_0^\infty m \cdot p_M(m) \cdot \left( F_X(\hat{x}(\ell) \mid M{=}m)] - \frac{1}{2}\left[ F_X(x \mid M{=}m) \right]_{\xi(\ell-1)}^{\xi(\ell)} \right) \mathrm{d}m = 0. \tag{7}$$

The zero of the left-hand-sided monotonous function in $\hat{x}(\ell)$ can be found using the bisection method in combination with numerical integration. Moreover, using the Monte-Carlo method, the centroid can be estimated as the weighted median (see Appendix D.3 (69))

$$\hat{x}(\ell) = \mathrm{median}_W(x_1, \ldots, x_{K_\ell}; w_1, \ldots, w_{K_\ell}) = \max_{\kappa \in \mathcal{K}_\ell} \left\{ x_\kappa \big| \sum_{k=1}^{\kappa} w_k \le \sum_{k=\kappa+1}^{K_\ell} w_k \right\}. \tag{8}$$

To constrain certain reconstruction levels during Lloyd's algorithm to specific values, e.g., -1, 0, 1, we initialize them with their predetermined values and skip their recomputation in each iteration. Note that for a fixed weight distribution, the EM algorithm only needs to be executed once, offline, and therefore adds no runtime cost during quantization of the network weights.

### 3.3 NOVEL OUTLIER-PRESERVING QUANTIZATION (OPQ)

Extreme outlier weights lead to suboptimal scaling of the associated block during block-wise absmax normalization. Therefore, block-wise quantization methods typically require small block sizes to limit the number of affected parameters. This increases the memory required to store the quantization constants. *To enable larger block sizes* and accordingly a smaller memory footprint, our outlier-preserving quantization (OPQ) approach stores outlier weights separately in `bfloat16` and additionally uses a 64-bit integer for each of them to address the outlier in the (flattened) weight tensor of the respective layer. We define outliers for each weight block independently as weights with an absolute value greater than the $q$-quantile of absolute block maxima after normalization of the block to a unit standard deviation. Formally, a weight $w_{b,i}$ is classified as an outlier if and only if

$$|w_{b,i}| > \sigma_b \cdot F_M^{-1}(q), \tag{9}$$

where $\sigma_b$ is the corrected sample standard deviation of the $b$-th block (see (73) in Appendix G), $F_M^{-1}()$ the quantile function of absolute block maxima (see (11) in Appendix D.1), and $q \in [0,1]$ is a hyperparameter controlling the number of affected outliers. Before quantization, we exclude outliers from the tensor by replacing them with zero, so that they are not considered in the subsequent (signed) block maximum search. Note that OPQ can be combined with either BOF4 or BOF4-S. For an in-depth explanation of our OPQ design choices, see Appendix G.

## 4 EXPERIMENTAL SETUP

In this section, we discuss our choices for the experimental evaluation of quantization methods.

**Quantized Models**: For evaluation, we apply quantization to three families of pre-trained LLMs: `Llama-3.1/3.2` (Dubey et al., 2024), `Qwen-2.5` (Yang et al., 2024), and `Mistral-7B-v0.3` (Jiang et al., 2023). By benchmarking across a diverse set of LLMs, we aim to demonstrate the generalizability of our method.

**Evaluated Quantization Methods**: We evaluate our proposed BOF4 and BOF4-S approaches, optimized w.r.t. either MAE or MSE. For the optimization, we always assume Gaussian network weights to ensure comparability to the baseline methods NF4 (Dettmers et al., 2023) and AF4 (Yoshida, 2023). However, we evaluate all approaches (of course) on real LLMs with weights, which are known to be only approximately Gaussian, see Appendix C. For the evaluation of OPQ, we performed a limited hyperparameter search, resulting in $q = 0.95$, see Appendix G.2. The optimized BOF4 and BOF4-S codebooks are provided in Appendix E.

**Fine-Tuning Method**: In addition to inference with quantization, we benchmark LLMs fine-tuned with quantization using the QLoRA method (Dettmers et al., 2023). The models are fine-tuned for instruction following using the Unnatural Instructions dataset (Honovich et al., 2023) or for code generation using the Magicoder-OSS-Instruct-75K dataset (Wei et al., 2024). Further details and hyperparameters can be found in Appendix H.

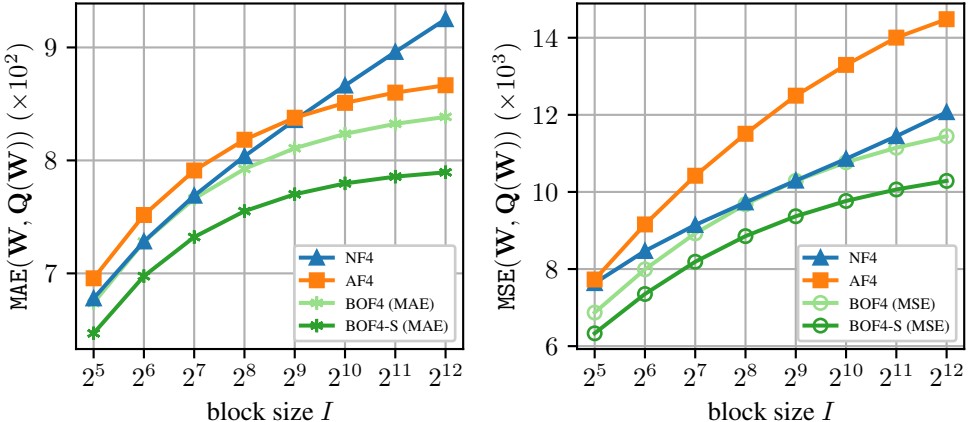

Figure 2: **MAE** (left) and **MSE** (right) **quantization error** of our quantization methods BOF4 and BOF4-S optimized for MAE (left, ∗) or MSE (right, ○) compared to the baselines NF4 and AF4 for Gaussian network weights $\mathbf{W} = (w_{b,i})$ with $w_{b,i} \sim \mathcal{N}(0, 1)$ depending on the block size $I$.

**Metrics**:   In order to show that our approach incurs reduced quantization errors, we report the mean squared error (MSE) and mean absolute error (MAE) of network weights. Following prior work (Frantar et al., 2023; Lin et al., 2024; Xiao et al., 2023), we assess the language modeling abilities of quantized models based mainly on the perplexity (PPL) measured on the WikiText-2 (Merity et al., 2017) and LAMBADA (Paperno et al., 2016) datasets. The perplexity on WikiText-2 is computed using the rolling log-likelihood with a maximum sequence length of 2048, as it is common in literature. Additionally, we evaluate the accuracy (ACC) in the NLP tasks MMLU (Hendrycks et al., 2021), ARC-Challenge (Clark et al., 2018), HellaSwag (Zellers et al., 2019), PIQA (Bisk et al., 2020), SIQA (Sap et al., 2019), and WinoGrande (Sakaguchi et al., 2021).

## 5   RESULTS AND DISCUSSION

**Quantization Error**:   In Fig. 2, we compare the MAE and MSE quantization errors of our proposed BOF4 and BOF4-S quantization methods with the baselines NF4 and AF4, assuming ideally Gaussian-distributed network weights. Accordingly, the results shown are independent of any particular LLM the methods are applied to. All compared quantizers constrain reconstruction levels such that 0 and the weight of the largest absolute value in a block are quantized without error, or in full 16-bit resolution, respectively. The error is computed empirically based on $2^{25}$ samples.

We observe that all investigated block-wise quantizers show increasing MAE / MSE with increasing block size $I$. This is expected, as larger block sizes will have larger block maxima, which in turn increases average error for the many non-maximum weights in the block. All of our proposed methods BOF4(-S), optimized w.r.t. both MAE and MSE, are equal to or better than each of the two baselines NF4 and AF4. Note that AF4 (Yoshida, 2023) was presented in some MAE-optimized form, which explains its poor MSE performance for medium- or large-sized blocks. *Our signed normalization method BOF4-S achieves lower MAE and MSE scores than any other investigated quantization approach.*

**Quantization Error and Perplexity**:   Tab. 1 shows a comparison of the MAE and MSE quantization errors, as well as perplexity (PPL), evaluated on the weights of three pre-trained LLMs: `Llama-3.1 8B`, `Qwen-2.5 7B`, and `Mistral 7B`. Results for additional (smaller) models are provided in Appendix J. Note that our intention is not to compare PPL between the various LLMs, but rather between the various quantizer options.

We observe that our basic BOF4 approaches are equal to or lower in quantization error than the baselines NF4 and AF4 when optimized for the particular metric MAE / MSE. The respective methods with *signed* normalization (BOF4-S) clearly outperform the non-signed BOF4 approaches in all cases,

Table 1: **Quantization error** (MAE and MSE) and **perplexity** (PPL) on WikiText-2 of quantization methods applied to the network weights of three LLMs with block size $I = 64$. Best result in each column in bold, second best underlined.

| | Llama-3.1 8B | | | Qwen-2.5 7B | | | Mistral 7B | | |
|---|---|---|---|---|---|---|---|---|---|
| | MAE↓ 1e−3 | MSE↓ 1e−6 | PPL↓ | MAE↓ 1e−4 | MSE↓ 1e−8 | PPL↓ | MAE↓ 1e−3 | MSE↓ 1e−6 | PPL↓ |
| NF4 | 0.977 | 1.637 | 8.53 | 1.202 | 2.391 | 9.89 | 2.256 | 8.439 | 8.90 |
| AF4 | 1.006 | 1.762 | 8.51 | 1.234 | 2.562 | 9.91 | 2.324 | 9.085 | 8.90 |
| BOF4 (MAE) | 0.976 | 1.621 | 8.52 | 1.202 | 2.370 | 9.89 | 2.256 | 8.360 | 8.90 |
| BOF4 (MSE) | 0.994 | 1.566 | 8.51 | 1.228 | 2.310 | 9.94 | 2.296 | 8.075 | 8.89 |
| BOF4-S (MAE) | 0.936 | 1.508 | 8.49 | 1.152 | 2.204 | 9.87 | 2.162 | 7.777 | 8.90 |
| + OPQ | **0.918** | 1.457 | 8.46 | **1.121** | 2.101 | **9.82** | **2.121** | 7.514 | 8.89 |
| BOF4-S (MSE) | 0.954 | 1.441 | 8.46 | 1.179 | 2.126 | 9.88 | 2.204 | 7.430 | 8.88 |
| + OPQ | 0.932 | **1.367** | **8.43** | 1.140 | **1.981** | 9.83 | 2.153 | **7.052** | **8.87** |

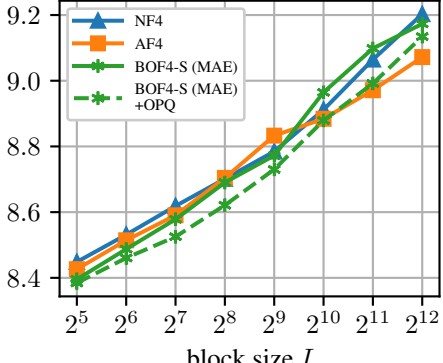 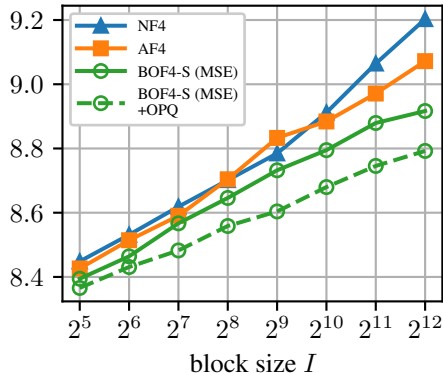

Figure 3: **Perplexity** of `Llama-3.1 8B` on WikiText-2 after quantization with **NF4**, **AF4**, and our **BOF4-S** optimized w.r.t. MAE (left, ∗) or MSE (right, ○) for different block sizes $I$, with and without outlier-preserving quantization (OPQ, dashed line).

and accordingly, the baselines NF4 and AF4 as well. We emphasize that MAE- and MSE-optimized BOF4(-S) schemes show the lowest quantization errors for their respective optimization metric. This empirically confirms our derived centroid update rules (7) and (5), along with the underlying Gaussian distribution assumption of the LLM network weights. Analyzing perplexity, BOF4-S is equal to (in a single case) or better than each of the baselines, indicating that the lower quantization error also pays off in terms of an improved language modeling accuracy. *Our proposed outlier-preserving quantization (OPQ) variant provides a further consistent performance improvement, as it lowers MAE and MSE quantization errors and perplexity in all cases.*

**Comparative Effect of MAE and MSE Optimization**: Tab. 1 also shows the language modeling perplexity of the quantization methods on the WikiText-2 dataset (Merity et al., 2017). This allows us to compare the effectiveness of BOF4(-S) optimized for MAE and MSE. We report both error metrics (MAE, MSE) for the quantized model weights w.r.t. the original model weights.

We observe the tendency of MSE-optimized BOF4(-S) methods to yield better (i.e., lower) perplexity than the MAE-optimized version, with only `Qwen-2.5 7B` being an exception with a 0.01 point perplexity advantage for MAE optimization. *Overall, the best-performing of our proposed schemes is BOF4-S (MSE) with OPQ, as it ranks either first or second among all other investigated methods in each metric.*

Fig. 3 shows the perplexity of `Llama-3 8B` on the WikiText-2 (Merity et al., 2017) and LAMBADA (Paperno et al., 2016) datasets after quantization with NF4, AF4, and our BOF4-S optimized w.r.t. MAE (left) and MSE (right). Furthermore, Fig. 3 reports the effect of utilizing the proposed outlier-

Table 2: **Inference** results of 4-bit scalar quantization methods evaluated using multiple LLMs with block size $I = 64$. The evaluated metrics are the perplexity on the WikiText-2 and LAMBADA dataset, and the accuracy on the MMLU (few-shot), ARC-Challenge, HellaSwag, PIQA, SIQA, and WinoGrande benchmarks. Best result in each column in bold, second best underlined, BF16 excluded.

| Model | Quantizer | WikiText2 PPL ↓ | Lambada PPL ↓ | MMLU ACC ↑ | ARC-C ACC ↑ | HellaSwag ACC ↑ | PIQA ACC ↑ | SIQA ACC ↑ | WinoGrande ACC ↑ | NAV ACC ↑ |
|---|---|---|---|---|---|---|---|---|---|---|
| Llama-3.2 3B | BF16 | 10.12 | 4.90 | 54.0 | 42.4 | 55.3 | 76.7 | 47.2 | 69.0 | 35.7 |
| | NF4 | 10.72 | 5.45 | 52.3 | 41.0 | **54.4** | 76.3 | 47.0 | 68.3 | 34.4 |
| | AF4 | 10.74 | 5.51 | 52.8 | 40.5 | **54.4** | 76.6 | **47.4** | 69.3 | **35.0** |
| | BOF4 (MSE) | 10.73 | 5.35 | 52.6 | **42.1** | 54.0 | 76.7 | 46.4 | 68.8 | 34.8 |
| | + OPQ | 10.67 | **5.17** | **52.9** | **42.1** | 54.1 | **76.9** | 46.4 | 68.4 | 34.8 |
| | BOF4-S (MSE) | 10.67 | 5.32 | 52.6 | 42.0 | 54.3 | 76.1 | 46.3 | 68.5 | 34.5 |
| | + OPQ | **10.64** | 5.25 | 52.5 | 41.8 | 54.2 | 76.2 | 46.5 | **69.5** | 34.9 |
| Qwen-2.5 3B | BF16 | 12.42 | 5.91 | 65.1 | 44.6 | 55.0 | 78.1 | 49.6 | 68.5 | 39.5 |
| | NF4 | 12.36 | 7.16 | 63.0 | 43.1 | 53.6 | 77.7 | **50.8** | 67.2 | 38.2 |
| | AF4 | 13.08 | 6.82 | 63.3 | 43.5 | **54.2** | **78.1** | 50.6 | 68.5 | 38.9 |
| | BOF4 (MSE) | 12.46 | 6.84 | **63.5** | 46.2 | 53.8 | 77.7 | 49.8 | 68.5 | 39.2 |
| | + OPQ | 12.48 | 6.90 | 63.1 | 46.2 | 54.1 | 77.5 | 50.2 | 67.6 | 38.9 |
| | BOF4-S (MSE) | 12.50 | 6.53 | **63.5** | 46.5 | 53.8 | 77.3 | 50.0 | 68.2 | 39.1 |
| | + OPQ | **12.35** | **6.43** | **63.5** | **46.6** | 54.0 | 77.5 | **50.8** | **69.1** | **39.7** |

preserving quantization (OPQ) in combination with BOF4-S. A corresponding figure including our BOF4 is given in Fig. 13 in Appendix J.

Fig. 3 (left) shows that our MAE-optimized BOF4-S methods reveal a lower PPL than both baselines up to block sizes of $I \leq 2^9$. The MAE-optimized baseline AF4 shows some strengths for very large block sizes $I \geq 2^{11}$, which, however, are not practically relevant.. When comparing to Fig. 3 (right), we observe that our MSE-optimized BOF4-S methods generally achieve a lower perplexity than both baselines and also than their MAE-optimized counterparts on the left. This trend becomes even more pronounced with increasing block size $I$. The overall better performance of our *MSE-optimized* BOF4(-S) approaches leads us to focus on these in the following experiments.

**Comparison to NF4 and AF4 for Inference**:    Tab. 2 shows the perplexity and accuracy of various quantized LLMs in the 3B regime on common NLP benchmarks. In addition, a *normalized average* accuracy (NAV ACC) is computed that accounts for the chance-level accuracy in each benchmark; for details about this metric, see Appendix K. Results for smaller and larger models are provided in Appendix J. Furthermore in Appendix I we provide additional inference evaluations demonstrating an application of our methods to calibration-data-based GPTQ quantization (Frantar et al., 2023).

Analyzing accuracy over the various benchmarks reveals that rank orders of models can be quite different in different benchmarks. Accordingly, such accuracy results should be interpreted with care. Our normalized average accuracy metric (last column) helps in identifying overall trends. For the Llama-3.2 3B model we see only slightly varying NAV ACC results, with AF4 and our BOF4-S +OPQ approach being close-by on first and second rank. On Qwen-2.5 3B, our favored BOF4-S +OPQ method has the overall best NAV ACC, even outperforming the BF16 reference. As we hardly claim to be better than 16 bit weight representation, we note once more the variance in the accuracy metric in general. *Among the two benchmarks reporting perplexity, our proposed BOF4-S +OPQ method ranks three times first and one time second, outperforming both baselines, NF4 and AF4..* In addition, OPQ incurs only a small runtime overhead during inference, as shown in Appendix G.3.

Note that the perplexity advantage of BOF4-S (+OPQ) was achieved despite the only partly valid Gaussian weight assumption in the design of our codebook. Our method also supports codebook optimization w.r.t. better-fitting distributions, which might even further improve performance.

**Fine-Tuning with Quantization**:    Tables 3 and 4 show the results for quantized fine-tuning using QLoRA (Dettmers et al., 2023) with various quantizers. Llama-3.2 3B is fine-tuned for instruction following and code generation, respectively, and evaluated on corresponding task-specific benchmarks. For comparison, we apply LoRA fine-tuning (Hu et al., 2022) to the original, unquantized weights in

Table 3: Prompt-level and instruction-level **accuracy** (%) on IFEval after **fine-tuning** `Llama-3.2 3B` for **instruction following** using 4-bit quantization with block size $I = 64$.

| | Prompt-level ACC ↑ | Instr.-level ACC ↑ | AVG ACC ↑ |
|---|---|---|---|
| Base Model | 21.1 | 33.6 | 27.3 |
| BF16 | 23.5 | 34.2 | 28.8 |
| NF4 | 24.4 | 35.0 | 29.7 |
| AF4 | 23.3 | 34.1 | 28.7 |
| BOF4 (MSE) | **26.8** | **36.5** | **31.6** |
| +OPQ | 25.0 | 34.8 | 29.9 |
| BOF4-S (MSE) | 24.4 | 35.1 | 29.8 |
| + OPQ | 25.0 | 35.0 | 30.0 |

Table 4: **Accuracy** (%) on the HumanEval+ and MBPP+ benchmarks after **fine-tuning** `Llama-3.2 3B` for **code generation** using 4-bit quantization with block size $I = 64$.

| | MBPP+ ACC ↑ | HumanEval+ ACC ↑ | AVG ACC ↑ |
|---|---|---|---|
| Base Model | 34.9 | 17.1 | 26.0 |
| BF16 | 37.8 | 30.5 | 34.2 |
| NF4 | 34.1 | 24.4 | 29.3 |
| AF4 | 32.8 | 23.2 | 28.0 |
| BOF4 (MSE) | 34.4 | 24.4 | 29.4 |
| +OPQ | 35.4 | 24.4 | 29.9 |
| BOF4-S (MSE) | 35.7 | 26.2 | 31.0 |
| + OPQ | **36.5** | **27.4** | **32.0** |

`bfloat16` representation (BF16). We find fine-tuning to be stable with all quantization methods. In addition to the task-specific accuracy metrics in Tables 3 and 4, we also report the average accuracy (AVG ACC).

*From a bird's-eye view over both tasks (tables) we observe the strength of BOF4-S +OPQ being confirmed*: For instruction following, it ranks second in AVG ACC, and for code generation, it ranks first—in both cases being better than NF4 and AF4.

Table 3, interestingly, reports our BOF4 approach as by far the best for instruction following. The OPQ variant for this particular downstream task is not the best. Accordingly, we keep in mind that for fine-tuning towards a specific task it might be advised to investigate which of our four proposed MSE-optimized BOF4 quantizers (signed vs. unsigned, with or without OPQ) performs best.

In Table 4, we observe for the code generation task that the previously best BOF4 is still equal to or better than the NF4 and AF4 baselines. The other three of our BOF4 variants are, however, even better in this case, with BOF4-S +OPQ being clearly ahead of all investigated approaches. The second rank is clearly taken by BOF4-S without OPQ. *This again confirms the recommendation that BOF4-based quantization of fine-tuned LLMs is best done after a small ablation study among the four MSE-optimized BOF4 quantizers.* For limitations of our work, see Appendix A.

## 6 CONCLUSIONS

In this paper, we analyzed block-wise absmax quantization for large language models (LLMs) and derived an expectation-maximization algorithm to minimize the quantization error. The resulting family of quantizers, termed 4-bit block-wise optimal float (BOF4), reduces the weight quantization error over previously published block-wise absmax quantizers such as NF4 (Dettmers et al., 2023) and AF4 (Yoshida, 2023). We also presented an improvement to the normalization technique by normalizing blocks of weights using their *signed* absolute maximum rather than the absolute maximum, which further reduces the quantization error and empirically mitigates the negative effect of quantization on perplexity. Our experimental study confirmed the importance of precisely representing zero and outlier network weights, and found that optimization w.r.t. the mean squared error (MSE) criterion results in lower perplexity compared to mean absolute error (MAE) optimization. Finally, we introduced outlier-preserving quantization (OPQ), a mixed-precision strategy for block-wise absmax quantization, which yields a significant perplexity advantage, especially at larger block sizes. We find that our methods can outperform NF4 and AF4 not only for inference, but also when used for fine-tuning with quantization, achieving higher accuracy on the target tasks.

Overall, our proposed methods can enable improved fine-tuning and inference for LLMs on consumer-grade hardware by boosting performance without increasing the memory footprint, thereby facilitating broader participation in both the scientific investigation and the application of LLMs.

REPRODUCIBILITY STATEMENT

For the sake of reproducibility, we make the source code for fine-tuning and inference evaluations, as well as codebook optimization, available at https://github.com/ifnspaml/bof4. In Appendix E, we provide some of the quantization codebooks resulting from our optimization algorithm. Additional codebooks are also provided in the GitHub repository. For our theoretical results, we state all assumptions and provide a detailed derivation in Appendix D.

ACKNOWLEDGMENTS

This work was partially funded by the German Federal Ministry for Research, Technology and Aeronautics (Bundesministerium für Forschung, Technologie und Raumfahrt, BMFTR) under the KI4ALL project (funding code: 16DHBKI055).

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

APPENDIX

## A  LIMITATIONS

Our evaluation focuses on comparisons with data-free quantization techniques and does not include post-training quantization (PTQ) methods that rely on calibration data. While quantization based on calibration data typically achieves better accuracy, we believe that data-free techniques are valuable because they are significantly more efficient in terms of time and memory required for quantizing the weights. Note that our contributions are compatible with and can be effectively applied to calibration-data-based PTQ methods, as demonstrated in Appendix I. Furthermore, the accuracy results of the utilized language modeling benchmarks may not be sensitive enough to reflect minor differences between quantization methods. We partially mitigate this issue by computing a normalized average accuracy score and relying more on perplexity as our primary metric. While we choose to evaluate our method to optimize quantization under the assumption of a Gaussian weight distribution to ensure a fair comparison with NF4 and AF4, our algorithm does not rely on a particular distribution. When applying double quantization as proposed by Dettmers et al. (2023), i.e., additionally quantizing the quantization constants beyond BF16, signed normalization would require an extra bit per block to encode the sign, raising memory consumption from 4.127 to 4.143 bits per weight. Without this, the improvement in the quantization error by using BOF4-S may be slightly diminished, since the input range of the quantizer for the quantization constants is doubled.

In future work, we want to combine our approach with Hadamard transformations of network weights used by recent quantization methods to ensure a Gaussian weight distribution (Ashkboos et al., 2024; Malinovskii et al., 2025). Furthermore, we would like to investigate in more detail how our contributions are best applied to calibration-data-based PTQ.

## B  ABLATION ON CONSTRAINED (I.E., FIXED) RECONSTRUCTION LEVELS

In Tab. 5, we evaluate the importance of precisely representing zero weights and absolute block maxima. We use the term "precise" in this context for an error-free representation of a zero weight and for a 16-bit representation of absolute block maxima. We measure the perplexity on WikiText-2 (Merity et al., 2017) of `Llama-3.1-8B` quantized with BOF4 for all four possible combinations of fixed (i.e., constrained) reconstruction levels $\hat{x}(\ell)$ from 0 and $\pm 1$.

Table 5: The **quantization error** (MAE and MSE) and **perplexity** (PPL) on WikiText-2 of **BOF4** with block size $I = 64$ using different combinations of fixed reconstruction levels applied to the network weights of `Llama-3.1-8B`. Best result in each column in bold.

| Constrained reconstruction levels | MAE↓ (1e−4) | MSE↓ (1e−6) | PPL↓ |
|---|---|---|---|
| $\varnothing$ | **9.881** | **1.506** | 8.81 |
| $\{0\}$ | 9.904 | 1.516 | 8.57 |
| $\{1, -1\}$ | 9.914 | 1.555 | 8.78 |
| $\{0, 1, -1\}$ | 9.936 | 1.566 | **8.51** |

We observe that fixing all (-1, 0, and 1) yields the best performance w.r.t. perplexity, even though the additional constraints on the codebook inevitably increase the quantization error. This confirms that the design choice of NF4 (Dettmers et al., 2023) and AF4 (Yoshida, 2023) to include these values as reconstruction levels is sound.

## C  WEIGHT DISTRIBUTION OF LARGE LANGUAGE MODELS

Dettmers et al. (2023) analyze the distribution of weight rows in a 7B LLaMa model to support the assumption of a Gaussian weight distribution for quantization. They perform a Shapiro-Wilk test on the rows and find that only 7.5% are non-Gaussian-distributed. We conduct a similar analysis for `Llama-3.1 3B` and `Qwen-2.5 3B` to evaluate whether this assumption holds for the models

used in this work. Instead of entire weight rows, we use blocks, since these are the units to which the quantizer is applied, and because the number of elements per row often exceeds the sample size at which the Shapiro-Wilk test produces reliable p-values. Using block size $I = 64$ and a significance threshold of 0.05 for the p-value, the test only identifies 6.9% and 10.5% of blocks as non-Gaussian for `Llama-3.1 8B` and `Qwen-2.5 8B`, respectively. When outliers are filtered prior to the test, as in OPQ (according to (9)), the fraction of non-Gaussian blocks decreases further to 4.6% and 6.5%, respectively. This indicates that assuming Gaussian-distributed weights for codebook optimization is reasonable for a large majority of the quantized blocks.

# D    FULL DERIVATION OF THE CORRECT OPTIMAL RECONSTRUCTION LEVELS

## D.1    DISTRIBUTION OF NORMALIZED WEIGHTS

To derive an optimized code for quantization, we first characterize the distribution of normalized weights. Yoshida performs a similar analysis, assuming zero-mean, unit-variance normally-distributed network weights $w_{b,i} \sim \mathcal{N}(0,1)$ (Yoshida, 2023). In the following, we generalize this analysis to weights distributed as *any symmetric, zero-mean probability distribution*. The weights $w_{b,i}$ are considered i.i.d. samples from a random variable $W$ with the probability density function (PDF) $p_W$ and cumulative distribution function (CDF) $F_W$. Similarly, $X$ is a random variable describing the distribution of the normalized weights with PDF $p_X$ and CDF $F_X$. A third random variable $M$ describes the distribution of absolute or signed block maxima $w_b^{\mathrm{max}}$. Fig. 4 shows an estimation of $p_X(x = x_{b,i})$ in case $w_{b,i} \sim p_W = \mathcal{N}(0,1)$. We observe that the distribution of normalized weights concentrates around zero with increasing block size $I$. Furthermore, the discrete fraction representing the absolute block maxima at the edges (-1 and 1) is inversely proportional to the block size $I$, as we will formalize later in (16).

**PDF for a Fixed Absolute Block Maximum:**    First, we analyze the distribution of normalized weights $X$ for a fixed absolute block maximum $M = m = w_b^{\mathrm{max}}$ (1). The PDF $p_X(x)$ with $x \in \mathbb{R}$ assigns a probability mass of $\frac{1}{2I}$ to both $x = -1$ and $x = +1$ since in the non-degenerated case there is *exactly one* value of maximum magnitude in each block of $I$ weights. The remaining probability mass of $\frac{I-1}{I}$ forms a continuous, non-uniform probability distribution on the interval $(-1, 1)$.

For a fixed absolute block maximum $m = w_b^{\mathrm{max}} \in \mathbb{R}^+$, the continuous portion of the CDF of $X$ is

$$
\begin{aligned}
F_X^{\mathrm{cont}}(x \mid M = m) &= P\big[X \le x \,\big|\, |X| < 1,\, M = m\big] \\
&= P\big[W < mx \,\big|\, |W| < m\big] \\
&= \frac{P[W \le mx \wedge |W| < m]}{P[|W| < m]} \\
&= \frac{F_W(mx) - F_W(-m)}{F_W(m) - F_W(-m)} \\
&= F_{W_{[-m,m]}}(mx),
\end{aligned}
\tag{10}
$$

for $x \in [0, 1]$, where $F_{W_{[-m,m]}}$ is the CDF $F_W$ of weights truncated to the interval $[-m, m]$.

**Distribution of Absolute Maxima:**    We continue with the distribution of the absolute block maxima, defined by the random variable $M$. First, due to the statistical independence of weights $w_{b,i}$ within a block, the CDF of $M$ is given by

$$
\begin{aligned}
F_M(m) &= P\big[|w_{b,1}| \le m, |w_{b,2}| \le m, \ldots, |w_{b,I}| \le m\big] \\
&= \prod_{i \in \mathcal{I}} F_{|W|}(m), \quad \mathcal{I} = \{1, 2, \ldots I\} \\
&= F_{|W|}^I(m),
\end{aligned}
\tag{11}
$$

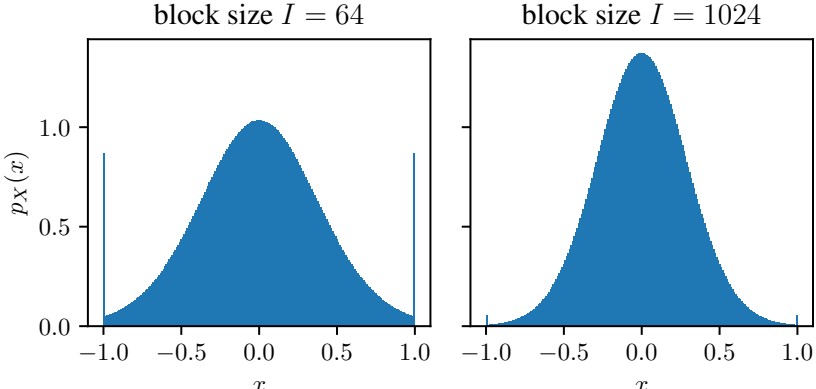

Figure 4: Empiric estimation of the PDF $p_X$, resulting from block-wise absmax normalization, in case of Gaussian network weights based on $2^{29}$ samples $x$ for different block sizes $I$.

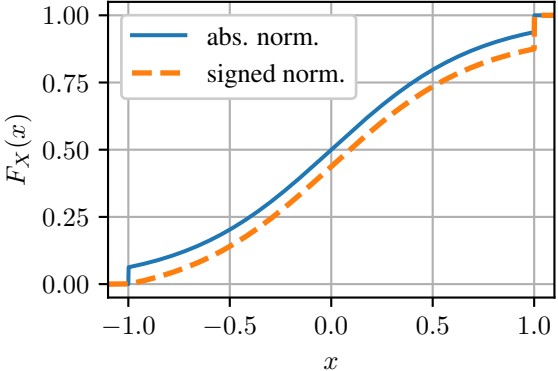

Figure 5: Example CDF $F_X(x)$ for absolute and signed block-wise absmax normalized Gaussian network weights $x = x_{b,i}$ and block size $I = 8$.

where $F_{|W|}$ is the CDF of $|W|$. Taking the derivative of $F_M$ using the chain rule, we obtain the PDF of $M$:

$$
\begin{aligned}
p_M(m) &= \frac{\mathrm{d}}{\mathrm{d}m} F_M(m) \\
&= I \cdot F_{|W|}^{I-1}(m) \cdot \frac{\mathrm{d}}{\mathrm{d}m} F_{|W|}(m) \\
&= I \cdot F_{|W|}^{I-1}(m) \cdot \frac{\mathrm{d}}{\mathrm{d}m}(2F_W(m) - 1) \\
&= 2I \cdot F_{|W|}^{I-1}(m) \cdot p_W(m)
\end{aligned}
\tag{12}
$$

In the derivation of (12), we use the fact that the PDF is the derivative of the CDF, and we use the equality

$$
\begin{aligned}
F_{|W|}(m) &= P[-m \leq W \leq m] \\
&= F_W(m) - (1 - F_W(m)) \\
&= 2F_W(m) - 1,
\end{aligned}
\tag{13}
$$

exploiting the symmetry of $p_W$ w.r.t. zero.

**CDF of the Normalized Weights:**  Now, the *continuous* part of the CDF $F_X(x)$, $-1 < x < 1$, can be calculated using the law of total probability by integrating over all possible values of the block-wise absolute maximum $M = m$, and weighting each block-maximum-dependent CDF with the corresponding probability density $p_M(m)$ as follows:

$$F_X^{\text{cont}}(x) = P\big[X_{b,i} \leq x \big| |X| < 1\big]$$
$$= \int_0^\infty p_M(m) \cdot F_X^{\text{cont}}(x \mid M{=}m) \, \mathrm{d}m. \tag{14}$$

By substituting the terms $p_M(m)$ and $F_X^{\text{cont}}(x \mid M{=}m)$ using (12) and (10), we obtain

$$F_X^{\text{cont}}(x) = 2I \cdot \int_0^\infty F_{|W|}^{I-1}(m) p_W(m) \cdot F_{W_{[-m,m]}}(mx) \, \mathrm{d}m. \tag{15}$$

Considering that $F_{X,\text{cont}}$ contains the fraction $\frac{I-1}{I}$ of the probability mass, whereas $+1$ and $-1$ each occur with probability $\frac{1}{2I}$, the CDF of $X$ can be characterized as

$$F_X(x) = \begin{cases} 0, & \text{if } x < -1 \\ \frac{1}{2I} + \frac{I-1}{I} F_X^{\text{cont}}(x), & \text{if } -1 \leq x < 1 \\ 1, & \text{if } x \geq 1 \end{cases}. \tag{16}$$

For signed block-wise absmax quantization, the continuous part of the distribution remains unchanged due to the symmetry of $p_W$ w.r.t. zero. Only the discrete probability mass of $1/I$ is now allocated entirely to $X = 1$, resulting in the CDF

$$F_X(x) = \begin{cases} 0, & \text{if } x < -1 \\ \frac{I-1}{I} F_X^{\text{cont}}(x), & \text{if } -1 \leq x < 1 \\ 1, & \text{if } x \geq 1 \end{cases}. \tag{17}$$

The CDF $F_X(x)$ for an *example* Gaussian weight distribution $w_{b,i} \sim p_W = \mathcal{N}(0,1)$ with block size $I = 8$ for both absolute and signed block-wise absmax quantization is shown in Fig. 5. We observe that, in the case of *absolute* block-wise absmax normalization, the CDF $F_X$ exhibits two discontinuities at $x = -1$ and $x = 1$, whereas, for *signed* absmax normalization, there is only one discontinuity at $x = 1$.

## D.2 Centroids for Block-Wise Normalized Weights

In the following, we show how Lloyd's algorithm (Lloyd, 1982) can be modified to minimize the $\text{MSE}(W, Q(W))$ or $\text{MAE}(W, Q(W))$ quantization error when using block-wise absmax quantization, where $Q : \mathbb{R} \to \mathbb{R}$ denotes the overall block-wise absmax quantization function. Lloyd's algorithm is applied to the normalized weights distributed according to $X$, whereas the quantization error should be minimized end-to-end for the distribution of original network weights $W$. We demonstrate how the centroid criterion for a reconstruction level $\hat{x}(\ell)$ of a region $\mathcal{R}_\ell = [\xi(\ell{-}1), \xi(\ell))$ can be reformulated to account for this discrepancy. Specifically, we derive a mathematical formula enabling the direct computation of the updated reconstruction level $\hat{x}(\ell)$ for any continuous distribution of network weights that is symmetrical w.r.t. zero and has a known PDF $p_W$ and CDF $F_W$. Furthermore, we consider the special case of Gaussian weights $p_W = \mathcal{N}(0,1)$, which allows further simplification in the case of the MSE criterion.

It should also be noted that the assignment of regions according to the nearest neighbor criterion remains unchanged from Lloyd's algorithm. The proof that the nearest neighbor criterion is still a necessary condition for optimality, even when applied to normalized weights, is trivial. Therefore, showing that our modified centroid criterion is a second necessary condition for optimality is sufficient to prove the local optimality of any solution to which our algorithm converges.

### D.2.1 MSE Optimization

First, we minimize the MSE quantization error $\text{MSE}(W, Q(W)) = \mathbb{E}_W[(W - Q(W))^2]$ with $\mathbb{E}_W[]$ being the expectation w.r.t. the weights $W$.

**General Centroid Criterion:** Our goal is to find a reconstruction level $\hat{x}(\ell)$ that minimizes the MSE quantization error for normalized weights that fall into region $\mathcal{R}_\ell$:

$$\hat{x}(\ell) = \underset{\hat{x} \in \mathbb{R}}{\arg\min} \, \mathrm{MSE}_\ell(W, Q(W)) = \underset{\hat{x} \in \mathbb{R}}{\arg\min} \, \mathbb{E}_W[(W - Q(W))^2 \mid X \in \mathcal{R}_\ell], \qquad (18)$$

where $\hat{x}$ represents a candidate value of the reconstruction level utilized for normalized weights falling into region $\mathcal{R}_\ell$ within $Q$, $\mathrm{MSE}_\ell(W, Q(W))$ is the MSE quantization error of weights that fall into region $\mathcal{R}_\ell$ after normalization.

Next, to enable analytical minimization of the reconstruction error, we use the law of total expectation

$$\mathbb{E}[A] = \mathbb{E}_B[\mathbb{E}_A[A \mid B]] = \int_{-\infty}^{\infty} \mathbb{E}[A \mid B = b] \cdot p(b) \, \mathrm{d}b. \qquad (19)$$

to express the expectation (18) as an integral over expectations conditioned on *a fixed block maximum* $M = m$:

$$\mathbb{E}_W[(W - Q(W))^2 \mid X \in \mathcal{R}_\ell]$$
$$= \int_0^{\infty} p_M(m \mid X \in \mathcal{R}_\ell) \cdot \mathbb{E}_W[(W - Q(W))^2 \mid M = m, X \in \mathcal{R}_\ell] \, \mathrm{d}m. \qquad (20)$$

This reformulation allows us to express the expectation in terms of the random variable $X$ and the sought reconstruction level $\hat{x}$:

$$\mathbb{E}_W\big[(W - Q(W))^2 \mid M = m, X \in \mathcal{R}_\ell\big]$$
$$= \mathbb{E}_X\big[(m \cdot X - m \cdot \hat{x})^2 \mid M = m, X \in \mathcal{R}_\ell\big]$$
$$= m^2 \cdot \mathbb{E}_X\big[(X - \hat{x})^2 \mid M = m, X \in \mathcal{R}_\ell\big] \qquad (21)$$

Substituting (21) into (20), we obtain the new formulation of an MSE-optimal reconstruction level

$$\hat{x}(\ell) = \underset{\hat{x} \in \mathbb{R}}{\arg\min} \int_0^{\infty} p_M(m \mid X \in \mathcal{R}_\ell) \cdot m^2 \cdot \mathbb{E}_X\big[(X - \hat{x})^2 \mid M = m, X \in \mathcal{R}_\ell\big] \, \mathrm{d}m. \qquad (22)$$

To find the optimal reconstruction level $\hat{x}$, we set the derivative w.r.t. $\hat{x}$ equal to zero:

$$\frac{\mathrm{d}}{\mathrm{d}\hat{x}} \int_0^{\infty} p_M(m \mid X \in \mathcal{R}_\ell) \cdot m^2 \cdot \mathbb{E}_X\big[(X - \hat{x})^2 \mid M = m, X \in \mathcal{R}_\ell\big] \, \mathrm{d}m$$
$$= \int_0^{\infty} m^2 \cdot 2\big(\hat{x} - \mathbb{E}_X[X \mid M = m, X \in \mathcal{R}_\ell]\big) \cdot p_M(m \mid X \in \mathcal{R}_\ell) \, \mathrm{d}m = 0 \qquad (23)$$

Rearranging for $\hat{x}$ yields

$$\hat{x}(\ell) = \hat{x} = \frac{\int_0^{\infty} m^2 \cdot \mathbb{E}_X[X \mid M = m, X \in \mathcal{R}_\ell] \cdot p_M(m \mid X \in \mathcal{R}_\ell) \, \mathrm{d}m}{\int_0^{\infty} m^2 \cdot p_M(m \mid X \in \mathcal{R}_\ell) \, \mathrm{d}m}. \qquad (24)$$

To compute this, we must express all quantities in terms of the known PDF $p_W$ and CDF $F_W$. To accomplish this, we analyze the PDF $p_M(m \mid X \in \mathcal{R}_\ell)$ of the weight maximum conditioned on the region $\mathcal{R}_\ell$. Using Bayes' theorem, we can express this as

$$p_M(m \mid X \in \mathcal{R}_\ell) = \frac{p_M(m) \cdot P[X \in \mathcal{R}_\ell \mid M = m]}{P[X \in \mathcal{R}_\ell]}$$
$$= \frac{p_M(m) \cdot \big(F_X(\xi(\ell) \mid M = m) - F_X(\xi(\ell-1) \mid M = m)\big)}{P[X \in \mathcal{R}_\ell]} \qquad (25)$$

The PDF of the weight maximum $p_M(m)$ is known from (12), and the CDF $F_X(x \mid M = m)$ for the continuous part of the distribution is known from (10). The special case of the non-continuous outermost regions is considered separately. With this and (24), the updated reconstruction level can be expressed as (major analytical result for MSE-optimized codebook reconstruction levels)

$$\boxed{\hat{x}(\ell) = \frac{\int_0^{\infty} m^2 \cdot \mathbb{E}_X[X \mid M = m, X \in \mathcal{R}_\ell] \cdot p_M(m) \cdot \big[F_X(x \mid M = m)\big]_{\xi(\ell-1)}^{\xi(\ell)} \, \mathrm{d}m}{\int_0^{\infty} m^2 \cdot p_M(m) \cdot \big[F_X(x \mid M = m)\big]_{\xi(\ell-1)}^{\xi(\ell)} \, \mathrm{d}m}.} \qquad (26)$$

Here, we use the notation $[F_X(x \mid M=m)]_{\xi(\ell-1)}^{\xi(\ell)} = F_X(\xi(\ell) \mid M=m) - F_X(\xi(\ell-1) \mid M=m)$. Using the known PDF $p_W$ and CDF $F_W$, $p_M(m)$ is obtained by (12), while $F_X(x \mid M=m)$ is obtained by (10). Hence, only the conditional expectation $\mathbb{E}_X[X \mid M=m, X \in \mathcal{R}_\ell]$ of normalized weights $X$ requires further analysis.

**Expectation of Normalized Weights:** To compute the optimal reconstruction level $\hat{x}(\ell)$ according to (26), we require a method to determine the expected value of the normalized weights, conditioned on the absolute block maximum $M = m$ and the region $\mathcal{R}_\ell = [\xi(\ell-1), \xi(\ell))$. First, we analyze $\mathbb{E}_X[X \mid M=m, X \in \mathcal{R}_\ell]$ under the assumption that the region is contained in the continuous part of the distribution $\mathcal{R}_\ell \subset (-1, 1)$. The conditional mean is given by

$$\mathbb{E}_X[X \mid M=m, X \in \mathcal{R}_\ell] = \frac{\int_{\mathcal{R}_\ell} x p_X(x \mid M=m) \, \mathrm{d}x}{\int_{\mathcal{R}_\ell} p_X(x \mid M=m) \, \mathrm{d}x}$$

$$= \frac{\int_{\mathcal{R}_\ell} x p_X(x \mid M=m)}{F_X(\xi(\ell) \mid M=m) - F_X(\xi(\ell-1) \mid M=m)} \tag{27}$$

We know from (10) that $F_X(x \mid M=m)$ for the continuous part of the distribution $\mathcal{R}_\ell \in (-1, 1)$ can be expressed as $F_{W_{[-m,m]}}(mx)$ using the CDF of $W$ truncated to the interval $[-m, m]$. The derivative w.r.t. $x$ represents the corresponding PDF

$$p_X(x \mid M=m) = \frac{\mathrm{d}}{\mathrm{d}x} F_{W_{[-m,m]}}(mx) = m p_{W_{[-m,m]}}(mx), \tag{28}$$

where $p_{W_{[a,b]}}$ for $a, b \in \mathbb{R}$ and $a < b$ is the PDF $p_W$ truncated to the interval $[a, b]$, formally defined as

$$p_{W_{[a,b]}}(w) = \frac{p_W(w) \cdot I_{[a,b]}(w)}{F_W(b) - F_W(a)}, \tag{29}$$

where $I_{[a,b]} : \mathbb{R} \to \{0, 1\}$ is the indicator function with $I_{[a,b]}(w) = 1 \Leftrightarrow a < w < b$. Note that our assumption $\mathcal{R}_\ell \subset (-1, 1)$ implies $-1 < X < 1$, and therefore $I_{[-m,m]}(mx) = 1$. Using this observation and (29), we get

$$\mathbb{E}_X[X \mid M=m, X \in \mathcal{R}_\ell] = \frac{\int_{\mathcal{R}_\ell} m \cdot x \cdot p_{W_{[-m,m]}}(mx) \, \mathrm{d}x}{F_{W_{[-m,m]}}(m\xi(\ell)) - F_{W_{[-m,m]}}(m\xi(\ell-1))}. \tag{30}$$

This expression can be simplified, since $p_{W_{[-m,m]}}(mx)$ and $F_{W_{[-m,m]}}(mx)$, shown in (10), share the common denominator $F_W(m) - F_W(-m)$, yielding (to be used in (26))

$$\boxed{\mathbb{E}_X[X \mid M=m, X \in \mathcal{R}_\ell] = \frac{\int_{\mathcal{R}_\ell} m \cdot x \cdot p_W(mx) \, \mathrm{d}x}{F_W(m\xi(\ell)) - F_W(m\xi(\ell-1))}.} \tag{31}$$

**Simplified Solution for Gaussian Network Weights:** In the following, we adopt the common assumption that the network weights are distributed according to a zero-mean unit-variance Gaussian $p_W(w) = g(w) := \mathcal{N}(w; 0, 1)$, enabling further simplification. Furthermore, we denote the CDF of the zero-mean unit-variance Gaussian as $G(w) = F_W(w)$. Additionally, we utilize the solution to the indefinite integral (Owen, 1980, equation (101))

$$\int x g(mx) = -\frac{1}{m^2} g(mx) + C, \tag{32}$$

with $C \in \mathbb{R}$. By applying this solution to (31), we obtain

$$\mathbb{E}_X[X \mid M=m, X \in \mathcal{R}_\ell] = -\frac{g(m\xi(\ell)) - g(m\xi(\ell-1))}{m\big(G(m\xi(\ell)) - G(m\xi(\ell-1))\big)}. \tag{33}$$

Substituting this into the centroid criterion (26) results in

$$\hat{x}(\ell) = -\frac{\int_0^\infty m \cdot \frac{g(m\xi(\ell)) - g(m\xi(\ell-1))}{G(m\xi(\ell)) - G(m\xi(\ell-1))} \cdot p_M(m) \cdot [F_X(x \mid M=m)]_{\xi(\ell-1)}^{\xi(\ell)} \, \mathrm{d}m}{\int_0^\infty m^2 \cdot p_M(m) \cdot [F_X(x \mid M=m)]_{\xi(\ell-1)}^{\xi(\ell)} \, \mathrm{d}m} \tag{34}$$

$$\boxed{\hat{x}(\ell) = \frac{\int_0^\infty m \cdot [g(mx)]_{\xi(\ell-1)}^{\xi(\ell)} \cdot (2G(m)-1)^{I-2} \cdot g(m) \, \mathrm{d}m}{\int_0^\infty m^2 \cdot [G(mx)]_{\xi(\ell-1)}^{\xi(\ell)} \cdot (2G(m)-1)^{I-2} \cdot g(m) \, \mathrm{d}m},} \tag{35}$$

for $\mathcal{R}_\ell \in (-1, 1)$. The integrals can be solved using numerical integration.

**Expectation of the Outermost Reconstruction Levels:** Next, we consider the edge case, where the centroid of the outermost region is to be computed, e.g., $-1 < \xi(\ell-1) < 1 \leq \xi(\ell)$, for the rightmost region. In this case, we can decompose the overall expectation in (26) as follows

$$
\begin{aligned}
&\mathbb{E}_X[X \mid M=m, X \in \mathcal{R}_\ell] \\
&= \quad P[\xi(\ell-1) \leq X < 1 \mid M = m, X \in \mathcal{R}_\ell] \cdot \mathbb{E}[X \mid M=m, \xi(\ell-1) \leq X < 1] \\
&\qquad + P[X=1 \mid M = m, X \in \mathcal{R}_\ell] \cdot 1
\end{aligned}
\tag{36}
$$

The expectation $\mathbb{E}_X[X \mid M=m, X \in \mathcal{R}_\ell]$ is decomposed into the expectation over the continuous part $\mathbb{E}_X[X \mid M=m, \xi(\ell-1) \leq X < 1)$ and the complementary case $X = 1$, each weighted by their respective probability. For the continuous part, we can compute the expectation with (33). The fraction of the probability mass in $\mathcal{R}_\ell$ that is allocated to $X = 1$ can be computed as

$$
\begin{aligned}
&P[X=1 \mid M=m, X \in \mathcal{R}_\ell] \\
&= \frac{P[X = 1, X \in \mathcal{R}_\ell \mid M=m]}{P[X \in \mathcal{R}_\ell \mid M=m]} \\
&= \frac{P[X=1 \mid M=m]}{P[X \in \mathcal{R}_\ell \mid M=m]} \\
&= \frac{P[X=1 \mid M=m]}{P[\xi(\ell-1) < X < 1 \mid M=m] + P[X=1 \mid M=m]}
\end{aligned}
\tag{37}
$$

For block-wise absmax normalization, without signed absmax, a fraction of $\frac{1}{2I}$ of the probability mass is allocated to 1, whereas the $\frac{I-1}{I}$ is contained in the continuous part of the distribution. We can use the CDF $F_X^{\text{cont}}(x \mid M = m) = F_{W_{[-m,m]}}(mx)$ to find the fraction of the probability mass contained in the continuous part of $\mathcal{R}_\ell$:

$$
\begin{aligned}
&P[\xi(\ell-1) < X < 1 \mid M=m] \\
&= P[-1 < X < 1 \mid M=m] \cdot P[\xi(\ell-1) < X \mid -1 < X < 1, M=m] \\
&= \frac{I-1}{I} \cdot (1 - F_X^{\text{cont}}(\xi(\ell-1) \mid M=m)) \\
&= \frac{I-1}{I} \cdot F_X^{\text{cont}}(-\xi(\ell-1) \mid M=m),
\end{aligned}
\tag{38}
$$

utilizing the symmetry of $F_X^{\text{cont}}()$ w.r.t. zero in the final step. Concerning (37), we obtain

$$
\begin{aligned}
&P[X=1 \mid M=m, X \in \mathcal{R}_\ell] \\
&= \frac{\frac{1}{2I}}{\frac{I-1}{I}(1 - F_X^{\text{cont}}(\xi(\ell-1) \mid M=m)) + \frac{1}{2I}} \\
&= \frac{1}{2(I-1)F_X^{\text{cont}}(-\xi(\ell-1) \mid M=m) + 1},
\end{aligned}
\tag{39}
$$

where $F_X^{\text{cont}}(x \mid M = m) = F_{W_{[-m,m]}}(mx)$ is the CDF of $X$ on the continuous part of the distribution derived in (10). The derivation for the leftmost reconstruction level is symmetric. When using signed absmax normalization, a probability of $\frac{1}{I}$ is assigned to 1, and we obtain instead

$$
\begin{aligned}
&P[X=1 \mid M=m, X \in \mathcal{R}_\ell] \\
&= \frac{1}{(I-1)F_X^{\text{cont}}(-\xi(\ell-1) \mid M=m) + 1},
\end{aligned}
\tag{40}
$$

for the rightmost reconstruction level, while the leftmost reconstruction level requires no special treatment.

Additionally, for the centroid according to (18), we require $F_X(x \mid M=m)$. For the continuous part of the distribution ($-1 < X < 1$) the solution is provided by (10). Accounting for the probability mass fraction $\frac{1}{I}$ distributed to $-1$ and 1, we obtain for absolute block-wise absmax normalization:

$$
F_X(x \mid M=m) = \begin{cases} 0, & \text{if } x < -1 \\ \frac{1}{2I} + \frac{I-1}{I}F_X^{\text{cont}}(x \mid M=m), & \text{if } -1 \leq x < 1 \\ 1, & \text{if } x \geq 1, \end{cases}
\tag{41}
$$

and for signed block-wise absmax normalization:

$$F_X(x \mid M{=}m) = \begin{cases} 0, & \text{if } x < -1 \\ \frac{I-1}{I} F_X^{\text{cont}}(x \mid M{=}m), & \text{if } -1 \le x < 1 \\ 1, & \text{if } x \ge 1. \end{cases} \tag{42}$$

It should also be noted that the computation for the continuous part of the distribution with CDF $F_X^{\text{cont}}(x \mid M{=}m)$ is identical for both signed and absolute block-wise absmax normalization. This is because the distribution of normalized weights only differs in the probability mass assigned to each of the endpoints $-1$ and $1$. Moreover, these edge cases are typically not evaluated, since the outermost reconstruction levels are usually constrained to $-1$ and $1$ and are not updated during Lloyd's algorithm.

### D.2.2 MAE OPTIMIZATION

We derive a condition for the centroid that minimizes the MAE quantization error $\text{MAE}(W, Q(W))$. We show that the condition minimizes the MAE by reducing the problem to the well-known fact that the median minimizes the mean absolute deviation from a set of points. Beginning analogously to Section D.2.1, we arrive at the following criterion for optimality:

$$\hat{x}(\ell) = \arg\min_{\hat{x} \in \mathbb{R}} \int_0^\infty m \cdot \mathbb{E}_X\big[|X - \hat{x}| \mid M{=}m, \, X \in \mathcal{R}_\ell\big] \cdot p_M(m \mid X \in \mathcal{R}_\ell)\, \mathrm{d}m. \tag{43}$$

Thus, we define the objective function we aim to minimize as

$$g(\hat{x}) := \int_0^\infty m \cdot \mathbb{E}_X\big[|X - \hat{x}| \mid M{=}m, \, X \in \mathcal{R}_\ell\big] \cdot p_M(m \mid X \in \mathcal{R}_\ell)\, \mathrm{d}m \tag{44}$$

Using the definition of the expected value, we have

$$\mathbb{E}_X\big[|X - \hat{x}| \mid M{=}m, X \in \mathcal{R}_\ell\big] = \int_{\mathcal{R}_\ell} p_X(x \mid M{=}m, X \in \mathcal{R}_\ell) \cdot |\hat{x} - x|\, \mathrm{d}x. \tag{45}$$

so that

$$g(\hat{x}) = \int_0^\infty \int_{\mathcal{R}_\ell} m \cdot p_M(m \mid X \in \mathcal{R}_\ell) \cdot p_X(x \mid M{=}m, X \in \mathcal{R}_\ell) \cdot |\hat{x} - x|\, \mathrm{d}x\, \mathrm{d}m. \tag{46}$$

We swap the order of integration:

$$g(\hat{x}) = \int_{\mathcal{R}_\ell} \left[ \int_0^\infty m \cdot p_M(m \mid X \in \mathcal{R}_\ell) \cdot p_X(x \mid M{=}m, X \in \mathcal{R}_\ell)\, \mathrm{d}m \right] |\hat{x} - x|\, \mathrm{d}x. \tag{47}$$

Now, we define a "re-weighted" PDF

$$\tilde{p}(x) := \frac{\int_0^\infty m \cdot p_M(m \mid X \in \mathcal{R}_\ell) \cdot p_X(x \mid M{=}m, X \in \mathcal{R}_\ell)\, \mathrm{d}m}{\int_0^\infty m \cdot p_M(m \mid X \in \mathcal{R}_\ell)\, \mathrm{d}m} \tag{48}$$

with which the objective function can be rewritten as

$$g(\hat{x}) = \int_0^\infty m \cdot p_M(m \mid X \in \mathcal{R}_\ell)\, \mathrm{d}m \cdot \int_{\mathcal{R}_\ell} \tilde{p}(x) \cdot |\hat{x} - x|\, \mathrm{d}x. \tag{49}$$

Since we search for $\arg\min_{\hat{x}} g(\hat{x})$, we can ignore the first integral in (49) as it is only a constant factor, and our objective function becomes

$$g(\hat{x}) := \int_{\mathcal{R}_\ell} \tilde{p}(x) \cdot |\hat{x} - x|\, \mathrm{d}x. \tag{50}$$

It is a well-known fact that for any PDF $\tilde{p}()$, the expected absolute deviation

$$\hat{x} \mapsto \int_{-\infty}^{\infty} \tilde{p}(x) \cdot |\hat{x} - x| \, \mathrm{d}x \tag{51}$$

is minimized by the median of the distribution with PDF $\tilde{p}(x)$. That is, if $\hat{x} = \hat{x}(\ell)$ is the minimum of $g(\hat{x})$, then it satisfies

$$\int_{-\infty}^{\hat{x}(\ell)} \tilde{p}(x) \, \mathrm{d}x = \frac{1}{2} \int_{\mathcal{R}_\ell} \tilde{p}(x) \, \mathrm{d}x. \tag{52}$$

The denominator $\int_0^{\infty} m \cdot p_M(m \mid X \in \mathcal{R}_\ell) \, \mathrm{d}m$ in (48) can be eliminated from both sides of (52), allowing us to redefine $\tilde{p}$ as

$$\tilde{p}(x) := \int_0^{\infty} m \cdot p_M(m \mid X \in \mathcal{R}_\ell) \cdot p_X(x \mid M = m, X \in \mathcal{R}_\ell) \, \mathrm{d}m. \tag{53}$$

Now, substituting the definition of $\tilde{p}(x)$ into the left-hand side of (52), we have

$$\int_{-\infty}^{\hat{x}(\ell)} \tilde{p}(x) \, \mathrm{d}x = \int_{-\infty}^{\hat{x}(\ell)} \left[ \int_0^{\infty} m \cdot p_M(m \mid X \in \mathcal{R}_\ell) \cdot p_X(x \mid M = m, X \in \mathcal{R}_\ell) \, \mathrm{d}m \right] \mathrm{d}x.$$

$$= \int_0^{\infty} m \cdot p_M(m \mid X \in \mathcal{R}_\ell) \cdot \left[ \int_{-\infty}^{\hat{x}(\ell)} p_X(x \mid M = m, X \in \mathcal{R}_\ell) \, \mathrm{d}x \right] \mathrm{d}m. \tag{54}$$

The inner integral is the conditional CDF of $X$ given $M = m$ and $X \in \mathcal{R}_\ell$. Thus,

$$\int_{-\infty}^{\hat{x}(\ell)} \tilde{p}(x) \, \mathrm{d}x = \int_0^{\infty} p_M(m \mid X \in \mathcal{R}_\ell) \cdot m \cdot F_X(\hat{x}(\ell) \mid M = m, X \in \mathcal{R}_\ell) \, \mathrm{d}m. \tag{55}$$

Similarly, the total mass in $\mathcal{R}_\ell$, on the right-hand side of (52), is

$$\int_{\mathcal{R}_\ell} \tilde{p}(x) \, \mathrm{d}x = \int_0^{\infty} m \cdot p_M(m \mid X \in \mathcal{R}_\ell) \, \mathrm{d}m. \tag{56}$$

Substituting (55) and (56) into (52), we obtain the new condition for optimality:

$$\int_0^{\infty} m \cdot p_M(m \mid X \in \mathcal{R}_\ell) \cdot F_X(\hat{x}(\ell) \mid M = m, X \in \mathcal{R}_\ell) \, \mathrm{d}m$$

$$= \frac{1}{2} \int_0^{\infty} m \cdot p_M(m \mid X \in \mathcal{R}_\ell) \, \mathrm{d}m. \tag{57}$$

Rearranging yields

$$\int_0^{\infty} m \cdot p_M(m \mid X \in \mathcal{R}_\ell) \cdot \left( F_X(\hat{x}(\ell) \mid M = m, X \in \mathcal{R}_\ell) - \frac{1}{2} \right) \mathrm{d}m = 0. \tag{58}$$

Finally, we use the definition of the truncated CDF and the characterization of $p_M(m \mid X \in \mathcal{R}_\ell)$ from (25) to obtain

$$\boxed{\int_0^{\infty} m \cdot p_M(m) \cdot \left( F_X(\hat{x}(\ell) \mid M = m) - \frac{1}{2} \left[ F_X(x \mid M = m) \right]_{\xi(\ell-1)}^{\xi(\ell)} \right) \mathrm{d}m = 0.} \tag{59}$$

All expressions can be computed directly from the known PDF $p_W$ and CDF $F_W$ of $W$, using (12), (41), (42). Thus, we can use this equation in Lloyd's algorithm to determine the centroid by numerical integration in combination with some method for finding the root of the monotonous function in $\hat{x}(\ell)$ on the left-hand side of (59), such as the bisection method. (Burden & Faires, 2010, pp. 48 ff.)

### D.3 EMPIRICAL CENTROID COMPUTATION

In Section D.2, we have derived theoretical solutions for the centroid computation in block-wise absmax quantization. However, computing the resulting integrals numerically might suffer from precision issues preventing a straight-forward implementation. Therefore, we additionally provide a simpler method to compute the centroid $\hat{x}(\ell)$ of a region $\mathcal{R}_\ell$ using Monte-Carlo estimation based on samples drawn from the network weight distribution. This method has the additional advantage that it can be applied to empirically collected weights from existing pre-trained networks rather than an assumed parametric distribution.

We first sample weights $\mathbf{W} = (\mathbf{W}_b) = (w_{b,i}) \in \mathbb{R}^{B \times I}$ grouped into $B$ blocks each with $I$ weights from the distribution of network weights $p_W$. Then, we normalize the weights by their respective block maxima $w_b^{\max}$ obtaining the normalized weights $\mathbf{X} \in \mathbb{R}^{B \times I} = (\mathbf{X}_b) = (x_{b,i})$. The objective now becomes to minimize the quantization error of the sampled weights $J(\mathbf{W}, \mathbf{Q}(\mathbf{W}))$, where $J$ is either $\mathrm{MAE}()$ or $\mathrm{MSE}()$ and $\mathbf{Q}(\mathbf{W})$ is the result of applying the quantization function $Q_b()$ element-wise to each row $\mathbf{W}_b$ of $\mathbf{W}$. We apply Lloyd's algorithm based on empirical data to the generated weights with a modified centroid criterion, which is derived in the following for both MAE and MSE optimization.

**MSE Optimization**: Our first goal is to minimize the MSE of the network weights based on the sampled weights $\mathbf{W}$. We can express $\mathrm{MSE}(\mathbf{W}, \mathbf{Q}(\mathbf{W}))$ in terms of an MSE of the normalized weights as follows:

$$
\begin{aligned}
\mathrm{MSE}(\mathbf{W}, \mathbf{Q}(\mathbf{W})) &= \frac{1}{B \cdot I} \sum_{b \in \mathcal{B}} \sum_{i \in \mathcal{I}} (w_{b,i} - Q_b(w_{b,i}))^2 \\
&= \frac{1}{B \cdot I} \sum_{b \in \mathcal{B}} \sum_{i \in \mathcal{I}} (w_{b,i} - w_b^{\max} \cdot \tilde{Q}(x_{b,i}))^2 \\
&= \frac{1}{B \cdot I} \sum_{b \in \mathcal{B}} \sum_{i \in \mathcal{I}} (w_b^{\max})^2 \cdot (x_{b,i} - \tilde{Q}(x_{b,i}))^2 \\
&= \frac{1}{B} \sum_{b \in \mathcal{B}} (w_b^{\max})^2 \cdot \mathrm{MSE}(\mathbf{X}_b, \tilde{\mathbf{Q}}(\mathbf{X}_b)),
\end{aligned}
\tag{60}
$$

where $w_b^{\max}$ for the $b$th block $\mathbf{W}_b \in (w_{b,i})$ is computed according to (1), and $\tilde{Q}()$ is the block-independent quantization function (3) that is utilized to quantize normalized weights.

Next, we show how the centroid computation using Lloyd's algorithm must be modified to update the reconstruction level $\hat{x}(\ell)$ in a *specific interval* $[\xi(\ell-1), \xi(\ell))$ using empirical samples of normalized weights $\mathbf{X} = (x_{b,i})$, such that the MSE of the weights $\mathrm{MSE}_\ell(\mathbf{W}, \mathbf{Q}(\mathbf{W}))$ *for that interval* is minimized. Let $x_k$, $k \in \mathcal{K}_\ell = \{1, \ldots, K_\ell\}$, be those normalized weights $x_{b,i}$ that fall into the interval $[\xi(\ell-1), \xi(\ell))$ in the $b$th block, and $w_k = w_b^{\max}$ the absolute or signed block maximum corresponding to the normalized weight $x_k$. Using (60), we can conclude that the contribution of the $\ell$th interval to the overall MSE is

$$
\mathrm{MSE}_\ell(\mathbf{W}, \mathbf{Q}(\mathbf{W})) = \frac{1}{K_\ell} \sum_{k \in \mathcal{K}_\ell} (w_k)^2 \cdot (x_k - \hat{x})^2.
\tag{61}
$$

We minimize by computing the derivative w.r.t. $\hat{x}$ according to

$$
\frac{\mathrm{d}}{\mathrm{d}\hat{x}} \mathrm{MSE}_\ell(\mathbf{W}, \mathbf{Q}(\mathbf{W})) = -\frac{2}{K_\ell} \sum_{k \in \mathcal{K}_\ell} w_k^2 \cdot (x_k - \hat{x}),
\tag{62}
$$

and setting it equal to 0, allowing us to ignore the constant factor:

$$
0 = \sum_{k \in \mathcal{K}_\ell} w_k^2 \cdot (x_k - \hat{x}(\ell)) = \sum_{k \in \mathcal{K}_\ell} w_k^2 \cdot x_k - \sum_{k \in \mathcal{K}_\ell} w_k^2 \cdot \hat{x}
\tag{63}
$$

Rearranging for $\hat{x}$, we obtain the optimal reconstruction level in the $\ell$th interval as

$$
\hat{x}(\ell) = \hat{x} = \frac{\sum_{k \in \mathcal{K}_\ell} w_k^2 \cdot x_k}{\sum_{k \in \mathcal{K}_\ell} w_k^2},
\tag{64}
$$

which is the weighted mean of weights $x_k$ in the interval $[\xi(\ell-1), \xi(\ell))$, weighted by their corresponding squared absolute or signed block maxima $w_k^2$. Tab. 8 (discussed in Appendix E) further supports the equivalence of this Monte-Carlo method with the theoretical solution given in (5).

**MAE Optimization:** A similar derivation can be made for the optimization of the MAE. Given fixed normalized network weights $x_k$, $k \in \mathcal{K}_\ell$, with associated absolute block maxima $w_k$, $k \in \mathcal{K}_\ell$, we are searching for

$$x(\ell) = \arg\min_{\hat{x}} \ \mathrm{MAE}_\ell(\mathbf{W}, \mathbf{Q}(\mathbf{W})) = \arg\min_{\hat{x}} \sum_{k \in \mathcal{K}_\ell} w_k \cdot |x_k - \hat{x}| \qquad (65)$$

Therefore, we define the function we aim to minimize as

$$f(\hat{x}) \coloneqq \sum_{k \in \mathcal{K}_\ell} w_k \cdot |x_k - \hat{x}|. \qquad (66)$$

where $x_k$, $k \in \mathcal{K}_\ell$, are those normalized network weights contained in the interval $[\xi(\ell-1), \xi(\ell))$. Further, we assume, w.l.o.g. that the normalized network weights $x_k$ are in ascending order: $x_1 \leq x_2 \leq \ldots \leq x_{K_\ell}$.

Obviously, the minimum must satisfy $x_1 \leq f(\hat{x}) \leq x_{K_\ell}$. Consider two distinct, consecutive normalized weights $x_\kappa, x_{\kappa+1}$ with $x_\kappa \neq x_{\kappa+1}$. Let $x_\kappa \leq \hat{x} < \hat{x}+\epsilon \leq x_{\kappa+1}$ for some $\epsilon \in \mathbb{R}^+$. Now, we can show the monotonicity of $f()$ on the interval $(x_\kappa, x_{\kappa+1})$ as follows:

$$
\begin{aligned}
f(\hat{x} + \epsilon) &= \sum_{k=1}^{\kappa} w_k \cdot (\hat{x}+\epsilon-x_k) + \sum_{k=\kappa+1}^{K_\ell} w_k \cdot (x_k - (\hat{x}+\epsilon)) \\
&= \sum_{k=1}^{\kappa} w_k \cdot \epsilon + \sum_{k=1}^{\kappa} w_k \cdot (\hat{x}-x_k) \\
&\quad - \sum_{k=\kappa+1}^{K_\ell} w_k \cdot \epsilon + \sum_{k=\kappa+1}^{K_\ell} w_k \cdot (x_k-\hat{x}) \\
&= \Big(\sum_{k=1}^{\kappa} w_k - \sum_{k=\kappa+1}^{K_\ell} w_k\Big) \cdot \epsilon + f(\hat{x}) \qquad (67)
\end{aligned}
$$

Therefore, $f(\hat{x})$ is monotonously decreasing on the interval $(x_\kappa, x_{\kappa+1})$ if

$$\sum_{k=1}^{\kappa} w_k \leq \sum_{k=\kappa+1}^{K_\ell} w_k, \qquad (68)$$

and monotonously increasing otherwise. Let $\kappa^{\mathrm{med}}$ be the largest index $\kappa$ for which (68) holds. Then, $f(\hat{x})$ is monotonously decreasing for $\hat{x} < x_{\kappa^{\mathrm{med}}}$ and monotonously increasing for $\hat{x} > x_{\kappa^{\mathrm{med}}}$. Therefore, $f(\hat{x})$ must be minimal at $\hat{x} = x_{\kappa^{\mathrm{med}}}$. The point $x_{\kappa^{\mathrm{med}}}$ is known as the *weighted median* of $x_k$, $k \in \mathcal{K}_\ell$, with weights $w_k$.

In conclusion, the optimal reconstruction level $\hat{x}(\ell)$ under the MAE criterion during the iteration of Lloyd's algorithm for normalized network weights is computed as the weighted median

$$\hat{x}(\ell) = \hat{x} = \mathrm{median}_{\mathrm{W}}(x_1, \ldots, x_{K_\ell}; w_1, \ldots, w_{K_\ell}) \coloneqq \max_{\kappa \in \mathcal{K}} \Big\{ x_\kappa \Big| \sum_{k=1}^{\kappa} w_k \leq \sum_{k=\kappa+1}^{K_\ell} w_k \Big\}, \quad (69)$$

for $x_k$, $k \in \mathcal{K}_\ell$, in ascending order $x_1 \leq x_2 \leq \ldots \leq x_{K_\ell}$, *both* in the case of block-wise absolute and signed absmax normalization. In both cases, $w_k$ represents the weight with the largest *absolute* value in the block containing $x_k$.

# E  OPTIMAL QUANTIZATION CODEBOOKS

Tab. 6 and Tab. 7 display the codebooks that were computed using the EM algorithm outlined in Section 3.2. In Tab. 6, the reconstruction levels of BOF4 and BOF-S optimized w.r.t. both MAE and

Table 6: Reconstruction levels $\hat{x}(\ell)$ of **BOF4** and **BOF4-S** optimized w.r.t. MAE and MSE for **block size** $I = 64$.

| | BOF4 | | BOF4-S | |
| | MAE-opt. $\hat{x}(\ell)$ | MSE-opt. $\hat{x}(\ell)$ | MAE-opt. $\hat{x}(\ell)$ | MSE-opt. $\hat{x}(\ell)$ |
| $\ell$ | | | | |
|---|---|---|---|---|
| 1 | -1.0 | -1.0 | -0.8018798232078552 | -0.8568463921546936 |
| 2 | -0.7026305794715881 | -0.7535245418548584 | -0.6076051592826843 | -0.6692874431610107 |
| 3 | -0.5272703766822815 | -0.579203724861145 | -0.468828022480011 | -0.5235266089439392 |
| 4 | -0.3946738243103027 | -0.4385998845100403 | -0.3559602797031403 | -0.4004882574081421 |
| 5 | -0.2832144796848297 | -0.3167679905891418 | -0.2576169371604919 | -0.2910638153553009 |
| 6 | -0.1835313588380814 | -0.2059924453496933 | -0.1677481383085251 | -0.1900092959403992 |
| 7 | -0.090308666229248 | -0.1015387624502182 | -0.0827366262674332 | -0.0938529595732689 |
| 8 | 0.0 | 0.0 | 0.0 | 0.0 |
| 9 | 0.0789600014686584 | 0.0887245312333107 | 0.0789434835314751 | 0.0887671709060669 |
| 10 | 0.1598792523145676 | 0.1793769598007202 | 0.1597966849803925 | 0.1794802695512772 |
| 11 | 0.244986355304718 | 0.2741499841213226 | 0.2448495477437973 | 0.2743096053600311 |
| 12 | 0.3372218906879425 | 0.3758211433887482 | 0.3371480107307434 | 0.3760197460651398 |
| 13 | 0.441359281539917 | 0.4884937703609467 | 0.4412573873996735 | 0.4886530041694641 |
| 14 | 0.565777063369751 | 0.6187058687210083 | 0.5656819343566895 | 0.6188603639602661 |
| 15 | 0.7299178242683411 | 0.7790452241897583 | 0.7298068404197693 | 0.7791395783424377 |
| 16 | 1.0 | 1.0 | 1.0 | 1.0 |

Table 7: Reconstruction levels $\hat{x}(\ell)$ of **BOF4-S** optimized w.r.t. MSE for **various block sizes** $I$.

| | $\hat{x}(\ell)$ for BOF4-S (MSE) | | | |
| $\ell$ | $I = 32$ | $I = 64$ | $I = 128$ | $I = 256$ |
|---|---|---|---|---|
| 1 | -0.8732797503471375 | -0.8568463921546936 | -0.83739173412323 | -0.8146829009056091 |
| 2 | -0.6907446384429932 | -0.6692874431610107 | -0.6462452411651611 | -0.6221838593482971 |
| 3 | -0.5437039136886597 | -0.5235266089439392 | -0.5028634667396545 | -0.4820549190044403 |
| 4 | -0.4173701703548431 | -0.4004882574081421 | -0.3836247622966766 | -0.3669650852680206 |
| 5 | -0.3038933575153351 | -0.2910638153553009 | -0.2783779501914978 | -0.2659871876239777 |
| 6 | -0.1986017823219299 | -0.1900092959403992 | -0.1815713942050934 | -0.1733742356300354 |
| 7 | -0.0981557220220566 | -0.0938529595732689 | -0.0896477326750755 | -0.0855776593089104 |
| 8 | 0.0 | 0.0 | 0.0 | 0.0 |
| 9 | 0.0925938412547112 | 0.0887671709060669 | 0.0850915610790253 | 0.0815095230937004 |
| 10 | 0.187048003077507 | 0.1794802695512772 | 0.1720834821462631 | 0.1649149656295776 |
| 11 | 0.2855197489261627 | 0.2743096053600311 | 0.2632072865962982 | 0.2524392008781433 |
| 12 | 0.3907126188278198 | 0.3760197460651398 | 0.3613293170928955 | 0.3470274209976196 |
| 13 | 0.506283164024353 | 0.4886530041694641 | 0.4707452654838562 | 0.4531534314155579 |
| 14 | 0.6379748582839966 | 0.6188603639602661 | 0.5988966822624207 | 0.578848659992218 |
| 15 | 0.7956376671791077 | 0.7791395783424377 | 0.761027991771698 | 0.7418596744537354 |
| 16 | 1.0 | 1.0 | 1.0 | 1.0 |

MSE are shown for an example block size $I = 64$. Tab. 7 shows the reconstruction levels of our top-performing quantizer BOF4-S optimized w.r.t. MSE for additional practical block sizes $I \leq 256$.

Furthermore, Tab. 8 presents a comparison of the BOF4 (MSE) reconstruction levels computed with two different implementations. In the first solution, the centroid is computed based on an empirical approach by the Monte-Carlo method using Gaussian-distributed data according to (6), while the second (theoretical) solution is computed data-independently using our implementation of (5) based on numerical integration. The variance in the finite number of Gaussian samples on the one hand, and numerical inaccuracies on the other hand, cause minor differences in reconstruction levels. The MSE between the theoretical and empirical solution is computed as (in dB)

$$\text{MSE} = 10 \cdot \log_{10} \frac{\sum_{\ell \in \mathcal{L}} P[X \in \mathcal{R}_\ell] \cdot \left( \hat{x}^{\text{theo}}(\ell) - \hat{x}^{\text{emp}}(\ell) \right)^2}{\sum_{\ell \in \mathcal{L}} P[X \in \mathcal{R}_\ell] \cdot \hat{x}^{\text{theo}}(\ell)^2} \text{ dB}, \tag{70}$$

Table 8: Reconstruction levels of **BOF4 (MSE)** for block size $I = 64$ using either the **empirical method** ($\hat{x}^{\mathrm{emp}}(\ell)$) or the **theoretical solution** ($\hat{x}^{\mathrm{theo}}(\ell)$) for the computation of centroids. The third column shows the absolute deviation between corresponding reconstruction levels.

| $\ell$ | Empirical solution $\hat{x}^{\mathrm{emp}}(\ell)$ | Theoretical Solution $\hat{x}^{\mathrm{theo}}(\ell)$ | Deviation $\|\hat{x}^{\mathrm{emp}}(\ell) - \hat{x}^{\mathrm{theo}}(\ell)\|$ |
|---|---|---|---|
| 1 | -1.0 | -1.0 | 0.0 |
| 2 | -0.7535245418548584 | -0.7535689203869577 | 0.0000443785320993 |
| 3 | -0.579203724861145 | -0.5792681492535123 | 0.0000644243923673 |
| 4 | -0.4385998845100403 | -0.4386720084478466 | 0.0000721239378063 |
| 5 | -0.3167679905891418 | -0.3168191039791481 | 0.0000511133900062 |
| 6 | -0.2059924453496933 | -0.2060291109696586 | 0.0000366656199653 |
| 7 | -0.1015387624502182 | -0.1015640796456471 | 0.0000253171954289 |
| 8 | 0.0 | 0.0 | 0.0 |
| 9 | 0.0887245312333107 | 0.0887646748673216 | 0.0000401436340109 |
| 10 | 0.1793769598007202 | 0.1794535266886747 | 0.0000765668879545 |
| 11 | 0.2741499841213226 | 0.274249773841407 | 0.0000997897200843 |
| 12 | 0.3758211433887482 | 0.375951029286045 | 0.0001298858972968 |
| 13 | 0.4884937703609467 | 0.4885925268369112 | 0.0000987564759645 |
| 14 | 0.6187058687210083 | 0.6187715546288008 | 0.0000656859077925 |
| 15 | 0.7790452241897583 | 0.7790828367844242 | 0.0000376125946659 |
| 16 | 1.0 | 1.0 | 0.0 |

where $\mathcal{L} = \{1, \ldots, 16\}$. With the results from Tab. 8, we obtain MSE $= -56.34\,\mathrm{dB}$. This demonstrates the *practical equivalence* of both implementations.

## F  OPTIMIZING THE QUANTIZATION ERROR OF NORMALIZED WEIGHTS

Instead of minimizing the end-to-end quantization error $\mathrm{MAE}(W, Q(W))$ or $\mathrm{MSE}(W, Q(W))$ of the network weights $w_{b,i}$ as in BOF4(-S), see Section 3.2, equations (7) and (5), one could alternatively minimize the quantization error $\mathrm{MAE}(X, Q_b(X))$ or $\mathrm{MSE}(X, Q_b(X))$ of the normalized weights $x_{b,i}$. In comparison to BOF4(-S), optimizing the quantization error of normalized weights is more straightforward and can be achieved using Lloyd's algorithm (Lloyd, 1982) with standard centroid update rules. For MAE minimization, the centroid of a Voronoi region $\mathcal{R}_\ell$ is computed based on samples from the network weight distribution with PDF $p_W$, as the median of normalized weights $x_k \in \mathbb{R}$, with $k \in \mathcal{K}_\ell = \{1, \ldots, K_\ell\}$:

$$\hat{x}(\ell) = \mathrm{median}(x_1, \ldots, x_{K_\ell}) \tag{71}$$

For MSE minimization, the optimal centroid is the mean

$$\hat{x}(\ell) = \frac{1}{K_\ell} \sum_{k \in \mathcal{K}_\ell} x_k. \tag{72}$$

Note that BOF4(-S) modifies these centroid conditions by introducing an additional weighting of the normalized network weights $x_k$ depending on the absolute block maxima $w_k$ of their respective block (see (8) for MAE and (6) for MSE).

We empirically compare the two optimization strategies. A 4-bit codebook minimizing $\mathrm{MSE}(X, Q_b(X))$ is computed with Lloyd's algorithm using centroids as defined in (72). Then, the perplexity of `Llama-3.1 8B` on WikiText-2 is measured for both this codebook (72) and BOF4 (MSE) (6). Figure 6 shows the difference in perplexity $\mathrm{PPL}_{\mathrm{BOF}} - \mathrm{PPL}_{\mathrm{NORM}}$ between the two optimization approaches, with $\mathrm{PPL}_{\mathrm{BOF}}$ referring to the perplexity achieved by BOF4 (MSE), and $\mathrm{PPL}_{\mathrm{NORM}}$ referring to the perplexity when using the codebook that minimizes $\mathrm{MSE}(X, Q_b(X))$. For all values of $I$, the difference is negative, indicating that BOF4 (MSE) consistently achieves lower perplexity than the codebook minimizing the MSE of normalized weights.

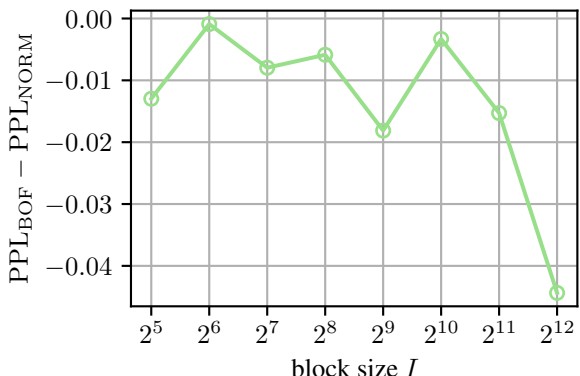

Figure 6: Difference in perplexity on WikiText-2 of `Llama-3.1 8B` quantized with BOF4 ($\text{PPL}_{\text{BOF}}$) vs. a codebook minimizing MSE of normalized weights ($\text{PPL}_{\text{NORM}}$). Lower values indicate better performance of BOF4.

## G  FURTHER DETAILS ON OUTLIER-PRESERVING QUANTIZATION (OPQ)

### G.1  DESIGN CONSIDERATIONS

We use a method to identify outliers that depends on the standard deviation $\sigma_b$ of weights within a block $b$ rather than on a fixed threshold, as the scaling of individual blocks within a neural network layer's weight tensor can vary greatly. Accordingly, we normalize the weights in each block to a standard deviation of 1, dividing by the sample estimate

$$\sigma_b = \sqrt{\frac{1}{I-1} \sum_{i \in \mathcal{I}} (w_{b,i} - \bar{w}_b)^2}, \quad b \in \mathcal{B}, \tag{73}$$

where $\bar{w}_b = \frac{1}{I} \sum_{i \in \mathcal{I}} w_{b,i}$ denotes the sample mean of weights $w_{b,i}$ in block $b$. Furthermore, to make the method generally applicable to different distributions $p_W$ of network weights, we use the expected distribution of absolute block maxima $w_b^{\max}$ ($p_M$ from (12)) to determine the threshold at which a normalized weight $w_{b,i}$ is classified as an outlier. Specifically, we use the $q$-quantile of the distribution of absolute block maxima with PDF $p_M$ for some value $q$ close to 1 as the threshold. Intuitively, this means that a normalized weight counts as an outlier if its absolute value is larger than a fraction $q$ of all absolute block maxima, assuming that the actual distribution of network weights would ideally adhere to our distribution assumption $p_W$.

Fig. 7 illustrates the detection of outliers. The blue histogram represents a block of absolute network weights $\frac{|w_{b,i}|}{\sigma_b}$, normalized by the standard deviation $\sigma_b$. The PDF $p_M$ (see (12)) describes the theoretical distribution of absolute block maxima, indicating where the largest absolute non-outlier weight is expected. In this example, we define outliers as absolute weights exceeding the 95th percentile of the expected absolute block maxima, denoted by $F_M^{-1}(0.95)$, the inverse of the CDF $F_M$ taken from (11). An example outlier is highlighted by red hatching.

Fig. 8 illustrates the advantage of applying OPQ to the network weights $\mathbf{W}$, which are almost Gaussian-distributed with only a small fraction of outlier weights that are highly unlikely to occur in Gaussian-distributed data. While OPQ stores the outlier weights (red color) in 16-bit precision, the non-outlier weights (blue color) are subject to normalization. On the right side in Fig. 8, the resulting normalized weights $\mathbf{X}$ without and with OPQ are shown for the weights that are no absolute block maxima, i.e., $x \in (-1, 1)$. Without applying OPQ, the outliers affect the scaling of their blocks during normalization, resulting in a distribution of normalized weights $\mathbf{X}$ that is more concentrated around the mean than the distribution $p_X^{\text{cont}}$ for which the quantizer was optimized. This is because during normalization, each block $b$ is divided by its absolute maximum $w_b^{\max}$. If $\mathbf{W}$ contains outliers, $w_b^{\max}$ is larger than expected for many blocks, leading to smaller normalized weights, which lets a quantizer operate in the underload regime, thereby being suboptimal w.r.t. its rate-distortion characteristics. On the other hand, when OPQ is used, the outlier weights are replaced with the placeholder value of 0 in

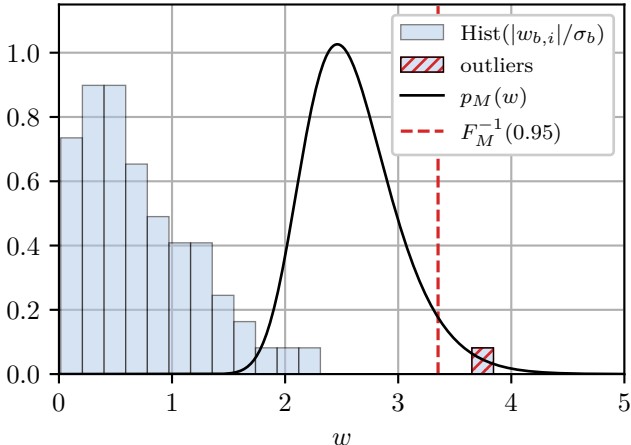

Figure 7: Illustration of **OPQ outlier detection**. The histogram of absolute weights $\frac{|w_{b,i}|}{\sigma_b}$ of an example block $b$ with block size $I = 64$ normalized to a unit standard deviation is shown in blue. Weights are identified as outliers (red hatching) iff they are greater than $F_M^{-1}(0.95)$, i.e., expected to be greater than $q = 95\%$ of the absolute block maxima $|w_b^{\max}|$ according to the assumption of Gaussian-distributed network weights $w_{b,i}$. The corresponding PDF $p_M$ of absolute block maxima $|w_b^{\max}|$ is shown as a black solid line.

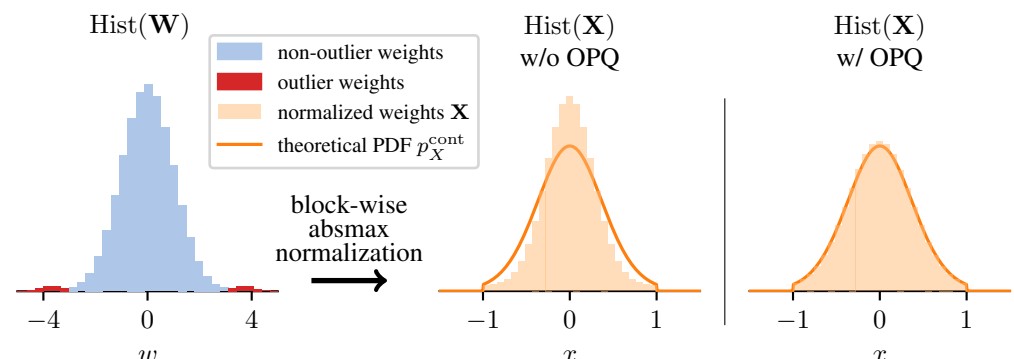

Figure 8: **Effect of outlier-preserving quantization** (OPQ) on the distribution of normalized network weights. The histogram of original network weights $\mathbf{W}$ containing some outlier weights (red) and non-outliers (blue) is shown on the left. The normalized network weights $\mathbf{X}$ that are not -1 or 1 are shown on the right with and without OPQ. The theoretical PDF $p_X^{\text{cont}}$ of the *continuous part* of normalized weights $\mathbf{X}$ is shown for comparison. The PDF $p_X^{\text{cont}}$ is computed under the assumption of Gaussian-distributed network weights, whereas the true network weights $\mathbf{W}$ contain (non-Gaussian) outliers.

the weight tensor $\mathbf{W}$ before normalization. Consequently, the distribution of normalized weights $\mathbf{X}$ is much more similar to the theoretically expected PDF $p_X^{\text{cont}}$. We chose this method for managing outliers, instead of abandoning the assumption of Gaussian network weights, because we observe that most rows of weight matrices in LLMs are very close to Gaussian, whereas only some blocks follow a super-Gaussian distribution with a small number of large-magnitude outlier weights. This observation is also supported by Dettmers et al. (2023, Appendix I). In practice, the design of OPQ enables one to control the expected number of weights stored in high precision via the choice of the hyperparameter $q$.

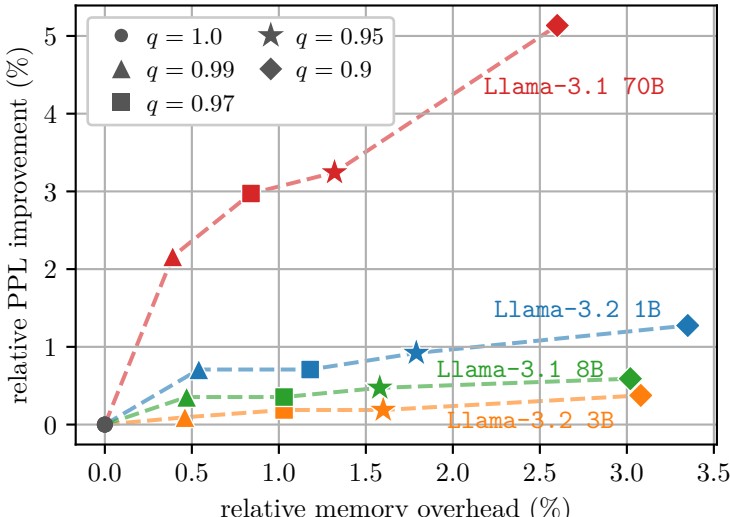

Figure 9: **Relative perplexity (PPL) improvement** on WikiText-2 validation split and **relative memory overhead** in relation to memory utilized by 4-bit quantized weights for **OPQ** at various model sizes and values of the hyperparameter $q$. All results use **BOF4-S** (MSE) with block size $I = 64$.

Note that the reconstruction levels of BOF4 or BOF4-S, shown in Tabs. 6 and 7, remain unchanged when OPQ is used.

### G.2 HYPERPARAMETER ABLATION

We perform a hyperparameter ablation to analyze the influence of the OPQ hyperparameter $q$ on the quantized model's performance and the additional memory cost compared to block-wise absmax quantization without OPQ. Fig. 9 presents the results for various Llama 3 models ranging from 1 billion to 70 billion parameters and hyperparameter choices $q \in \{0.9, 0.95, 0.97, 0.99, 1.0\}$. The relative perplexity improvement from OPQ is calculated as

$$\Delta\mathrm{PPL}(q) = \frac{\mathrm{PPL}_{\mathrm{ref}} - \mathrm{PPL}(q)}{\mathrm{PPL}_{\mathrm{ref}}} \tag{74}$$

where $\mathrm{PPL}(q)$ is the perplexity of the quantized model on the WikiText-2 (Merity et al., 2017) validation split using BOF4-S (MSE) with OPQ at a specific value for the hyperparameter $q$, and $\mathrm{PPL}_{\mathrm{ref}}$ is the reference perplexity resulting from BOF4-S (MSE) without OPQ. The memory overhead is calculated in relation to the memory usage by 4-bit quantized layers, excluding layers that are not quantized, such as the embedding and final layers.

We observe that OPQ yields a substantially larger relative perplexity improvement for the `Llama-3.1 70B` model compared to the other significantly smaller LLMs with $1 \ldots 8$ billion parameters. Notably, this improvement is achieved without any increase in relative memory overhead. This suggests that, in very large LLMs, preserving weight outliers is even more critical for maintaining performance. For a fixed value of $q$, the relative memory overhead is similar between model sizes. For instance, at $q = 0.95$ the overhead ranges from 1.32% and 1.79% for all evaluated models. In smaller LLMs, decreasing $q$ below 0.95 provides only marginal perplexity gains while increasing memory overhead. For very large models, however, choosing a lower value of $q < 0.95$ may be viable if the deployment hardware can accommodate the additional memory overhead, since perplexity continues to improve beyond this point. Based on these observations, we recommend $q = 0.95$ as a robust hyperparameter choice *across model sizes*.

Additionally, we analyze how the block size $I$ affects the choice of the hyperparameter $q$ by measuring the additional memory cost (Fig. 10) and the perplexity on the WikiText-2 validation split (Fig. 11), using `Llama-3.1 8B` as an example.

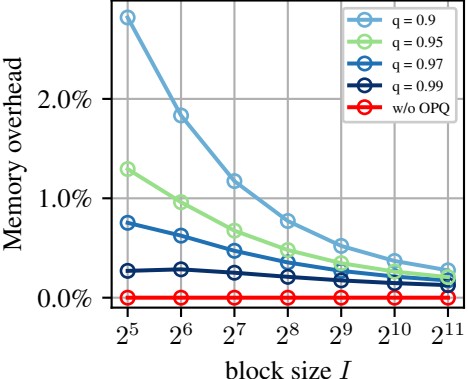 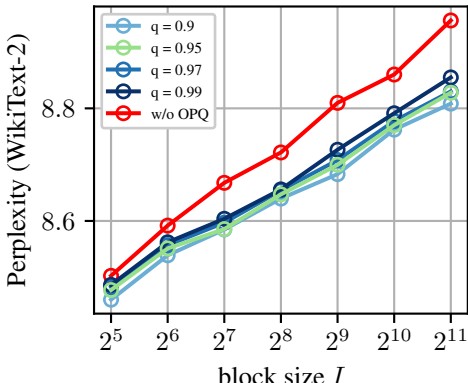

Figure 10: **Additional OPQ memory overhead** of **BOF4** (MSE) applied to `Llama-3.1 8B` as a fraction of the total memory required by the quantized model after block-wise absmax quantization, including the quantization constants.

Figure 11: **Perplexity** of `Llama-3.1 8B` on the WikiText-2 validation split after quantization with **BOF4** (MSE) using **OPQ** with various values of the hyperparameter $q$.

We observe that the memory overhead decreases as the block size $I$ increases. Moreover, the positive effect of OPQ on perplexity increases with increasing block size. For instance, when setting $q = 0.9$, the memory overhead at block size $I = 32$ is approximately $3\%$ while the effect on perplexity is low. Meanwhile, at larger block sizes, the effect on perplexity becomes more pronounced and OPQ only incurs a minimal memory overhead, even for $q = 0.9$. Across block sizes, we find that $q = 0.95$ (light-green curves) leads to an acceptable fraction of weights stored in high precision, even for small block sizes, and consistently improves perplexity at every tested block size. This confirms our earlier observation from Fig. 9 that $q = 0.95$ is a robust hyperparameter choice.

### G.3 RUNTIME OVERHEAD

We additionally evaluate the runtime overhead of OPQ. Figure 12 shows the time required to generate 1000 tokens with `Llama-3.1 8B` on an NVIDIA RTX 4070 Ti Super GPU using block-wise absmax quantization without and with OPQ. We use a batch size of 1 and start generating from an empty context. Note that the particular block-wise absmax quantization method that is used does not influence the decoding runtime, since NF4, AF4, BOF4, and BOF4-S all utilize the same implementation of decoding, only differing in the values of the reconstruction levels $\hat{x}(\ell)$. We measure a runtime overhead of $2.88\%$ on average for OPQ, which is nearly constant in the block size $I$, with the largest measured overhead being $3.84\%$. *This shows that OPQ only incurs a minimal runtime overhead.*

## H TRAINING AND EVALUATION DETAILS

Our hyperparameter choices align closely with those used by Dettmers et al. during the original evaluation of the QLoRA method (Dettmers et al., 2023). We use the AdamW optimizer (Loshchilov & Hutter, 2019) with a constant learning rate of $4 \cdot 10^{-5}$, configured with the exponential decay rates $\beta_1 = 0.9$ and $\beta_2 = 0.999$. We perform supervised fine-tuning for 1875 steps using batch size 16. Furthermore, we use gradient clipping with a `max_grad_norm` parameter of 0.3. A dropout with a $10\%$ dropout rate is applied to the LoRA layers. In contrast to Dettmers et al. (2023), we do not perform double qunaitzation, i.e., the quantization constants are not further quantized.

We use the LM Evaluation Harness by EleutherAI (Gao et al., 2024) (MIT License) for the evaluation on the NLP benchmarks. The code generation benchmarks HumanEval+ and MBPP+ are provided by the EvalPlus framework (Liu et al., 2023) (Apache 2.0 License).

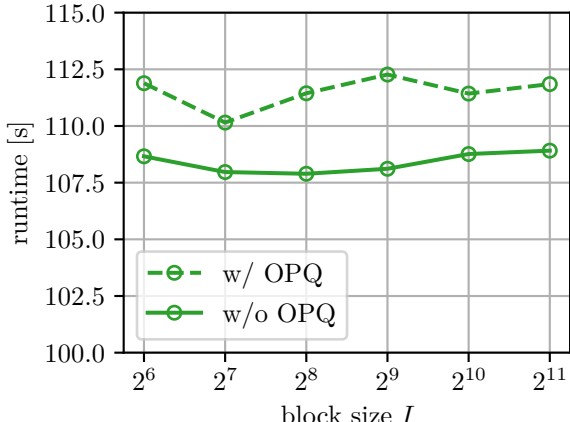

Figure 12: Time to generate 1000 tokens starting from an empty context with and without OPQ, depending on the block size $I$, evaluated on `Llama-3.1 8B` using a batch size of 1
.

All fine-tuning runs were conducted on a single A100 40GB GPU. Each run finished in less than 8 hours. For perplexity and accuracy evaluations, either an NVIDIA RTX 3080 with 10GB of memory or an A100 40GB was used. Each evaluation run of a (quantized) model on our set of NLP benchmarks required at most 24 hours.

## I    APPLICATION TO CALIBRATION-DATA-BASED QUANTIZATION

For most experiments, we evaluate BOF4 in a data-free setting, where quantization is determined solely by the network weights rather than by network activations computed with the help of calibration data. This ensures a fair comparison with the data-free methods NF4 and AF4, which we improve upon. While data-free quantization offers this comparability, post-training data-aware methods often achieve higher accuracy. Data-aware approaches, such as GPTQ (Frantar et al., 2023), AWQ (Lin et al., 2024), and SmoothQuant (Xiao et al., 2023), leverage calibration data to optimize the assignment of network weights to reconstruction levels. Most of these methods rely on uniform quantization, yet the uniform codebook can be readily replaced with a non-uniform one. In fact, the use of non-uniform codebooks in GPTQ has already been explored by van Baalen et al. (2024). Furthermore, GPTQ natively supports operating on normalized blocks of weights (referred to as groups by (Frantar et al., 2023)), making it a natural framework to evaluate our BOF4 method in a data-aware setting. Moreover, GPTQ can be trivially modified to use our signed block-wise normalization (Section 3.1) method and OPQ (Section 3.3).

As a proof of concept, we experimentally evaluate this approach. Frantar et al. (2023) demonstrate that GPTQ already works well with larger block sizes between 128 and 1024. Accordingly, we evaluate all methods at block size $I = 128$. As calibration data, we use 512 samples from the C4 dataset (Raffel et al., 2020).

The results are shown in Tab. 9. We observe that using GPTQ with non-uniform BOF4 (MSE) quantization significantly improves perplexity and average accuracy compared to the uniform quantization from the original GPTQ proposal. Further improvements are achieved by adding signed normalization (BOF4-S) and outlier-preserving quantization (OPQ). Overall, perplexity is reduced from 8.60 to 8.34 on WikiText2 and from 4.32 to 3.93 on Lambada, while normalized average accuracy improves from 42.0% to 42.8%. This significantly closes the gap to full-precision (BF16) inference. Furthermore, we observe that non-uniform quantization generally improves perplexity and accuracy significantly: non-uniform, *data-free* quantization performs better than uniform, *data-aware* GPTQ in both perplexity metrics and competitively in terms of accuracy. This highlights that non-uniform quantization is a viable option that should be considered in cases where memory is the main concern and improving inference speed is secondary. *Our results demonstrate that our*

Table 9: **Inference** with **GPTQ quantization** (Frantar et al., 2023) with block size $I = 128$ and various underlying codebooks evaluated using `Llama-3.1 8B`. Best-performing data-free methods are shown for comparison. The evaluated metrics are the perplexity on the WikiText-2 and LAMBADA dataset, and the accuracy on the MMLU (few-shot), ARC-Challenge, HellaSwag, PIQA, SIQA, and WinoGrande benchmarks. Best result in each column in bold, second best underlined, BF16 excluded.

| Quantizer | WikiText2 PPL ↓ | Lambada PPL ↓ | MMLU ACC ↑ | ARC-C ACC ↑ | HellaSwag ACC ↑ | PIQA ACC ↑ | SIQA ACC ↑ | WinoGrande ACC ↑ | NAV ACC ↑ |
|---|---|---|---|---|---|---|---|---|---|
| BF16 | 7.94 | 3.96 | 63.0 | 51.3 | 60.0 | 80.0 | 47.0 | 73.8 | 43.4 |
| BOF4-S (MSE) | 8.57 | 4.04 | 61.4 | 48.3 | 59.3 | 79.7 | 46.5 | 72.4 | 41.5 |
| +OPQ | 8.48 | 4.05 | 61.7 | 48.9 | 59.3 | **79.8** | 46.7 | 73.0 | 42.0 |
| GPTQ, uniform | 8.60 | 4.32 | 60.9 | 50.2 | 59.3 | 79.2 | 47.2 | 72.9 | 42.0 |
| GPTQ, BOF4 (MSE) | 8.41 | 3.96 | 62.0 | 49.9 | 59.3 | 79.2 | 47.4 | 73.0 | 42.3 |
| GPTQ, BOF4-S (MSE) | 8.36 | **3.92** | **62.3** | 49.8 | **59.5** | 79.0 | **48.1** | 72.5 | 42.3 |
| + OPQ | **8.34** | 3.93 | 62.2 | **50.5** | 59.3 | 79.7 | 47.3 | **73.7** | **42.8** |

*contributions—BOF4 codebook optimization, signed normalization, and OPQ—can be naturally incorporated into calibration-data-based quantization methods, yielding clear benefits.*

## J ADDITIONAL EVALUATIONS

In Tab. 10, the quantization error and perplexity results for the `Llama-3.1 8B` and the `Qwen-2.5 7B` models are shown. Note that Tab. 10 corresponds to Tab. 1, the latter just showing larger models. Similar to the larger models, we observe that our BOF4(-S) quantizers perform at least as well and usually better than the baseline methods in the quantization error metric (MAE or MSE) for which the particular codebook is optimized. Furthermore, the BOF4-S quantizers using our signed absmax normalization significantly improve all metrics over BOF4 with absolute absmax normalization. When additionally applying OPQ to the BOF4-S quantizers, performance in all metrics improves further. *The lowest errors are achieved by BOF4-S +OPQ, using the codebook optimized for the particular error metric.* Interestingly, for the `Qwen-2.5 3B` model, our MAE-optimized methods generally achieve better perplexity, suggesting that the target error metric for optimization, which leads to the best performance, may vary depending on the LLM.

Fig. 13 shows perplexity results for BOF4 on the WikiText-2 and LAMBADA datasets. Note that Fig. 13 corresponds to Fig. 3, the latter reporting on BOF4-S, however. *We observe that for most block sizes I, the perplexity of BOF4 optimized w.r.t. MAE or MSE is similar to that of the best-performing baseline method. Adding OPQ significantly reduces perplexity, particularly when used with MSE optimized codebooks and at large block sizes.*

Tab. 11 displays additional perplexity and accuracy measurements for the larger `Llama-3 8B`, `Qwen-2.5 7B`, and the tiny `Qwen-2.5 0.5B`. Note that Tab. 11 corresponds to Tab. 2, the latter reporting on small 3B models. Our best BOF4 quantization method consistently outperforms the baseline methods, NF4 and AF4, in terms of perplexity on WikiText-2 and LAMBADA, except when applied to the `Qwen-2.5 7B model`, where NF4 achieves a surprisingly low perplexity on LAMBADA—surpassing even the performance of the unquantized BF16 model. Note, however, that for the `Qwen-2.5 7B` model, each of our four proposed BOF4(-S) methods performs as well as or better than NF4 in the NLP benchmarks' normalized average accuracy (NAV) metric. Overall normalized average accuracy (NAV) results from the language modeling benchmarks do not indicate a single quantizer or approach that consistently performs best.

The OPQ variant proves particularly effective for the smaller `Qwen-2.5 0.5B` model, where it significantly improves perplexity over the respective quantizer without OPQ and both baselines NF4 and AF4.

Table 10: **Quantization error** (MAE and MSE) and **perplexity** (PPL) on WikiText-2 of quantization methods applied to the network weights of two 3B regime LLMs with block size $I = 64$. Best result in each column in bold, second best underlined.

| | Llama-3.1 3B | | | Qwen-2.5 3B | | |
|---|---|---|---|---|---|---|
| | MAE↓ 1e−3 | MSE↓ 1e−6 | PPL↓ | MAE↓ 1e−3 | MSE↓ 1e−6 | PPL↓ |
| NF4 | 1.399 | 3.333 | 10.72 | 1.822 | 5.722 | 12.13 |
| AF4 | 1.441 | 3.588 | 10.71 | 1.862 | 6.118 | 13.48 |
| BOF4 (MAE) | 1.399 | 3.302 | 10.72 | 1.821 | 5.670 | 12.16 |
| BOF4 (MSE) | 1.424 | 3.191 | 10.73 | 1.862 | 5.526 | 12.46 |
| BOF4-S (MAE) | 1.341 | 3.071 | 10.68 | 1.746 | 5.274 | 12.07 |
| + OPQ | **1.316** | 2.971 | **10.63** | **1.689** | 5.026 | **12.05** |
| BOF4-S (MSE) | 1.367 | 2.936 | 10.66 | 1.788 | 5.087 | 12.41 |
| + OPQ | 1.336 | **2.791** | 10.64 | 1.719 | **4.739** | 12.36 |

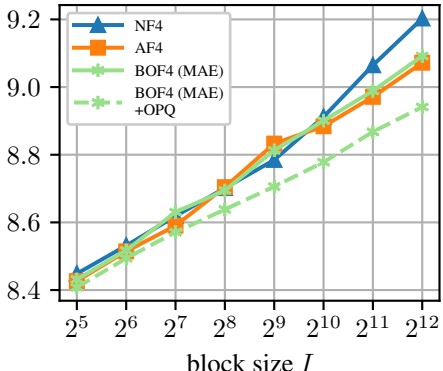 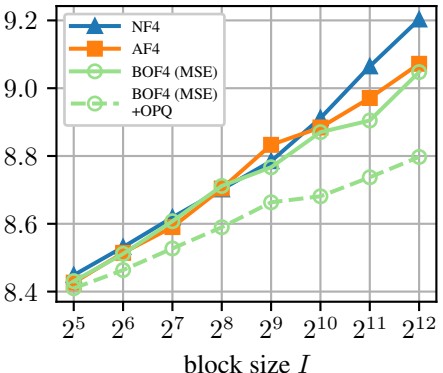

Figure 13: **Perplexity** of `Llama-3.1 8B` on WikiText-2 after quantization with **NF4**, **AF4**, and our **BOF4** optimized w.r.t. MAE (left, ∗) or MSE (right, ∘) for different block sizes $I$, without and with outlier-preserving quantization (OPQ, dashed line).

## K  DEFINITION OF THE NORMALIZED AVERAGE ACCURACY METRIC

To determine an overall accuracy score for a model over multiple benchmarks, we employ a normalized average accuracy that accounts for the chance level accuracy achievable by random guessing on each benchmark. For example, some benchmarks use a multiple-choice format with four answer options. In this case, random guessing would yield an accuracy of 25%. To ensure that no benchmark disproportionately influences the average accuracy, we normalize the accuracy of multiple-choice benchmarks such that random guessing is expected to yield 0% and answering all queries correctly yields 100%. The normalized accuracy is calculated as

$$\text{ACC}_{\text{norm}} = \frac{\text{ACC} - \text{ACC}_{\text{chance}}}{1 - \text{ACC}_{\text{chance}}}, \tag{75}$$

where $\text{ACC}_{\text{chance}}$ is the chance-level accuracy. This normalized average accuracy $\text{ACC}_{\text{norm}}$ is reported in Tabs. 2 and 11, abbreviated as NAV ACC.

Table 11: **Inference** results of 4-bit scalar quantization methods evaluated using **additional LLMs** with block size $I = 64$. The evaluated metrics are the perplexity on the WikiText-2 and LAMBADA dataset, and the accuracy on the MMLU (few-shot), ARC-Challenge, HellaSwag, PIQA, SIQA, and WinoGrande benchmarks. Best result in each column in bold, second best underlined, BF16 excluded.

| Model | Quantizer | WikiText2 PPL ↓ | Lambada PPL ↓ | MMLU ACC ↑ | ARC-C ACC ↑ | HellaSwag ACC ↑ | PIQA ACC ↑ | SIQA ACC ↑ | WinoGrande ACC ↑ | NAV ACC ↑ |
|---|---|---|---|---|---|---|---|---|---|---|
| Llama-3.1 8B | BF16 | 7.94 | 3.96 | 63.0 | 51.3 | 60.0 | 80.0 | 47.0 | 73.8 | 43.4 |
| | NF4 | 8.53 | 4.41 | 61.2 | 49.1 | 59.1 | 78.9 | **47.4** | **73.6** | 42.0 |
| | AF4 | 8.51 | 4.38 | 61.6 | 49.9 | 59.1 | 79.5 | 47.0 | **73.6** | **42.4** |
| | BOF4 (MSE) | 8.47 | **4.25** | 61.7 | **50.4** | 59.3 | 78.9 | 46.4 | 73.1 | 42.0 |
| | + OPQ | 8.47 | **4.25** | 61.7 | **50.4** | 59.3 | 78.9 | 46.4 | 73.1 | 42.0 |
| | BOF4-S (MSE) | 8.47 | 4.29 | 61.7 | 48.5 | **59.5** | 79.2 | 46.2 | 72.8 | 41.6 |
| | + OPQ | **8.43** | 4.29 | **61.9** | 49.2 | **59.5** | **79.7** | 46.5 | 72.5 | 41.9 |
| Qwen-2.5 7B | BF16 | 9.50 | 4.53 | 71.5 | 48.2 | 60.0 | 78.7 | 54.8 | 72.7 | 45.8 |
| | NF4 | 9.91 | **4.48** | 70.7 | 46.7 | 59.0 | **78.9** | 54.2 | 71.7 | 44.6 |
| | AF4 | 9.90 | 4.70 | 70.6 | 47.2 | 58.9 | 78.3 | **54.5** | 70.2 | 44.0 |
| | BOF4 (MSE) | 9.95 | 4.83 | 70.7 | 48.2 | 59.2 | 78.7 | 54.1 | 71.3 | 44.8 |
| | + OPQ | 9.85 | 4.73 | 70.6 | 47.4 | 59.2 | **78.9** | 54.2 | **72.2** | **45.0** |
| | BOF4-S (MSE) | 9.88 | 4.79 | **70.8** | 48.4 | 59.2 | 78.6 | 54.3 | 70.6 | 44.6 |
| | + OPQ | **9.83** | 4.67 | 70.6 | **48.5** | **59.3** | 78.6 | 54.4 | 71.1 | 44.8 |
| Qwen-2.5 0.5B | BF16 | 19.64 | 16.95 | 47.5 | 29.5 | 40.6 | 70.2 | 44.4 | 56.4 | 21.1 |
| | NF4 | 22.24 | 25.20 | 44.8 | 28.3 | 38.8 | **69.5** | **44.4** | 56.6 | **19.7** |
| | AF4 | 22.14 | 27.17 | 43.5 | 28.5 | 39.0 | 68.9 | 43.3 | **56.8** | 19.1 |
| | BOF4 (MSE) | 22.22 | 27.28 | **45.1** | 29.9 | **39.1** | **69.5** | 42.9 | 54.5 | 19.1 |
| | + OPQ | **21.72** | 24.61 | 45.0 | 29.0 | 39.0 | 69.4 | 43.5 | 55.6 | 19.3 |
| | BOF4-S (MSE) | 23.02 | 26.64 | 44.2 | **30.4** | **39.1** | 68.0 | 43.7 | 55.6 | 19.1 |
| | + OPQ | 21.88 | **22.90** | 44.2 | 29.6 | 38.8 | 68.4 | 43.4 | 56.7 | 19.2 |

