# OpenReview forum: "Improving Block-Wise LLM Quantization by 4-bit Block-Wise Optimal Float (BOF4): Analysis and Variations"
_ICLR.cc/2026/Conference — ICLR 2026 Poster_

### Official Review · Reviewer_SU8o · 2025-10-28

**Soundness:** 2
**Presentation:** 3
**Contribution:** 1
**Rating:** 2
**Confidence:** 4

**Summary:**

The paper proposes a new 4-bit data format called FOF4, which leverages the asymmetric property of maximum/minimum values within a block to reduce the waste of degrees of freedom in the quantization codebook. This method can be regarded as a type of non-uniform quantization.

**Strengths:**

This method leverages the asymmetry of the boundaries in weight quantization to reduce quantization error.

**Weaknesses:**

1. This method can be regarded as a special type of non-uniform quantization method (1D codebook). Additionally, compared with other KMeans-based methods such as GPTVQ and RPTQ, it may not have advantages in terms of accuracy and speed (without hardware support).
2. This method lacks a comparison with similar basic codebook-based methods like GPTVQ，VPTQ...
3. Current LLMs can achieve W4A4 quantization with almost no loss, which has greater advantages in reducing computational overhead. This method seems unable to achieve this, and it also requires additional unstructured outlier storage and computation.

**Questions:**

Methods that previously determined normalization constants based on MSE have also achieved good results. Why is the weight based on the maximum absolute value chosen instead? What advantages does this approach offer?

---

> ### Author Response · Authors · 2025-11-19
>
> We thank the reviewer for their constructive feedback. We answer their concerns and questions below.
>
> The reviewer’s assessment of our contribution (summary and strengths) appears to focus solely on our signed normalization approach, without acknowledging our other contributions—most notably, (1) the theoretically optimal codebook optimization algorithm and (2) the outlier-preserving quantization method. We encourage the reviewer to revisit our paper and to take these aspects into account when evaluating the scope of our work.
>
> **On weaknesses 2 and 3**: The reviewer argues that our method is not compared to other non-uniform LLM quantization methods like GPTQV and VPTQ. Furthermore, the reviewer notes that current state-of-the-art quantization methods can achieve near-lossless 4-bit quantization, in contrast to BOF4.
>
> We would like to point out that the mentioned methods (GPTQV and VPTQ) use multi-dimensional vector quantization and calibration data to assign codebook points to codebook vectors. Many other state-of-the-art methods also depend on calibration data. In contrast, we evaluate our method in the *scalar* and *data-free* setting, making a direct comparison to calibration-based methods quite unfair.
>
> In fact, Appendix I shows that our codebook optimization approach *can be combined with calibration-based methods* such as GPTQ, where it yields substantial improvements. In principle, our optimization approach can also be extended to multi-dimensional vector quantization, which we leave to future work. In this paper, we focus on deriving an algorithm for the computation of theoretically optimal codebooks for block-wise absmax quantization and point out mistakes in previous such attempts. Consequently, we use the previous efforts to construct optimal non-uniform codebooks (NF4 and AF4) as our baselines.
>
> We want to emphasize that quantization of weight blocks normalized by their absolute maximum is a common practice in LLM quantization, and our codebook optimization algorithm can potentially improve any such quantization scheme.
>
> **On weakness 1**: The reviewer also mentions that non-uniform quantization approaches usually do not yield an advantage in terms of speed. This is a valid point. However, in some deployment scenarios, the available memory is harshly limited and speed is not a primary concern. In these cases, non-uniform quantization excels because it can deliver superior accuracy at the same memory footprint. The results reported in Tab. 9 support this point by showing that our BOF4 codebook significantly outperforms uniform quantization when used in combination with GPTQ.
>
> The reviewer asked:
>
> > Methods that previously determined normalization constants based on MSE have also achieved good results. Why is the weight based on the maximum absolute value chosen instead? What advantages does this approach offer?
>
> Methods that determine normalization constants based on MSE can work well, but they do not preserve the largest-magnitude weight in each block. In Appendix B, we demonstrate the importance of accurately representing the absolute maximum value in a block. Using the (signed) absolute block maximum as normalization constant ensures that the largest-magnitude value in each block is reconstructed exactly, provided the codebook includes reconstruction levels at normalized weight values of -1 and 1. This preserves some of the salient outlier weights even without our *optional* OPQ method. In contrast, MSE would be minimized by optimizing the non-uniform codebook for the expected distribution of normalized weights without constraining reconstruction levels to -1 and 1. However, Tab. 5 shows that this approach—despite the improved MSE—yields worse perplexity than the codebook with constrained reconstruction levels. Another issue with computing MSE-optimal normalization constants is that it would introduce a bidirectional dependency between the choice of the codebook and the choice of optimal normalization constants, making it difficult to compute an optimal codebook for the distribution of normalized weights, which our algorithm achieves for (signed) absmax normalization.

---

### Official Review · Reviewer_dVvQ · 2025-10-29

**Soundness:** 3
**Presentation:** 4
**Contribution:** 3
**Rating:** 6
**Confidence:** 2

**Summary:**

This paper introduces a novel 4-bit block-wise quantization algorithm called BOF4-S. The proposed algorithm employs an enhanced EM algorithm to obtain the optimal reconstruction levels. Additionally, it normalizes the weights using the signed absolute block maximum, which effectively saves one reconstruction level, thereby adding an extra degree of freedom to the codebook and further reducing quantization error. Moreover, this algorithm excludes outlier weights and store them separately to further enhance the quantization accuracy. Experimental results demonstrate that the proposed algorithm achieves less quantization error and perplexity compared to the conventional methods, while also achieving superior model performance before and after fine-tuning.

**Strengths:**

1. This paper provides a thorough and clear explanation of the method, making it easy for readers to understand. The experimental results are comprehensive, which strongly supports the effectiveness of the proposed method.

2. The method addresses several practical issues in current quantization algorithms, such as the wastage of reconstruction levels and the impact of outliers on quantization accuracy. By cleverly saving reconstruction levels, designing an optimized codebook algorithm, and eliminating outliers, it provides an ingenious approach to optimizing quantization algorithms, further enhancing the quantization accuracy.

3. This method achieves improvements in both quantization error and perplexity with minimal time overhead. Furthermore, the LLMs quantized using this algorithm demonstrate better task performance before and after fine-tuning compared to traditional methods, showcasing the method's strong applicability.

**Weaknesses:**

Although selecting one of the two endpoints as the reconstruction level for the maximum absolute weight provides an additional degree of freedom for the codebooks, it also requires an extra bit to store the sign of this maximum value. Is this overhead justified? Especially since, when using Llama-3.2-3B as the base model, there is no performance improvement on most tasks (Table 2).

**Questions:**

1. Is it possible for both endpoints of the normalized weights (-1 and 1) to appear at the same time? If so, how is the sign handled?
2. When outliers are removed, their corresponding values in the tensor are replaced with 0. Is this tensor modified before or after normalization? Does directly replacing values with 0 impact the distribution of weights within the current block, potentially interfering with the selection of the optimal codebooks?

---

> ### Author Response · Authors · 2025-11-19
>
> We thank the reviewer for their thoughtful review. We are pleased that they rated the presentation as excellent and recognized our good contributions. Below, we address their concerns and questions.
>
> The reviewer mentions that our signed normalization method would require an extra bit to store the sign.
>
> We would like to clarify that an additional bit for the sign of the absolute maximum value is not required unless *double quantization* is used, in which the quantization constants are quantized again. When the quantization constants are stored in a typical 16-bit floating-point format (float16, bfloat16), there is no memory overhead, since these data types already support signed values. All of our experiments were performed without using double quantization, meaning that *no memory overhead was incurred for signed normalization*. We also note that only one additional bit *per block* would be required for double quantization. At the typical block size $I=64$, and using the same hyperparameters as Dettmers et al. [1] for double quantization, this would increase the used memory from 4.127 to 4.143 bits *per weight*, marking a 0.3\% increase in memory consumption. We believe that this small memory overhead is justified by the consistently improved language modeling perplexity from signed normalization that is shown in Tabs. 1 and 10.
>
> We address the reviewers’ questions below:
>
> > 1. Is it possible for both endpoints of the normalized weights ($-1$ and $1$) to appear at the same time? If so, how is the sign handled?
>
> Yes, this is possible, although it is rare in practice. For this to occur, two weights within the same block must have identical absolute values in their bfloat16 representation. In such cases, the implementation can choose either $-1$ or $+1$ as the quantization constant for that block—no special handling is required. The only effect is that one of the normalized weights will take on the value $-1$.
>
> > 2. When outliers are removed, their corresponding values in the tensor are replaced with 0. Is this tensor modified before or after normalization? Does directly replacing values with 0 impact the distribution of weights within the current block, potentially interfering with the selection of the optimal codebooks?
>
> The tensor is modified *before* normalization. In fact, the beneficial effect of this step on the distribution of normalized weights is a *key advantage* of our OPQ method, which is illustrated in Fig. 8. Since we define outliers as values that are significantly larger than the expected block maximum under the assumed weight distribution, filtering them before normalization aligns the distribution of normalized weights more closely with the assumed distribution. Therefore, it *reduces* the quantization error. The consistently lower MAE and MSE quantization errors with OPQ, reported in Tabs. 1 and 10, further support that OPQ effectively shapes the normalized weight distribution towards the Gaussian form assumed in the optimization of these particular codebooks. We would like to note that replacing outliers with zeros during quantization does not directly affect the codebook optimization algorithm, since the codebook is pre-optimized “offline” based on the assumed weight distribution.
>
> [1] Dettmers et al., “QLORA: Efficient Finetuning of Quantized LLMs”, 2023. Proc. of NIPS

---

### Official Review · Reviewer_sXWd · 2025-10-31

**Soundness:** 3
**Presentation:** 2
**Contribution:** 3
**Rating:** 6
**Confidence:** 4

**Summary:**

This paper presents a comprehensive study on improving 4-bit block-wise quantization for Large Language Models (LLMs). The authors identify a fundamental issue in existing methods (e.g., NF4, AF4): they optimize the quantization error of the *normalized* weights rather than that of the *original* weights, leading to suboptimal results. To address this, the authors propose an improved Expectation-Maximization (EM) algorithm that directly optimizes the error of the original weights, resulting in a family of quantizers termed BOF4. They also introduce signed absolute maximum normalization, which frees up one codebook entry to enhance representational capacity. Additionally, a mixed-precision scheme is proposed to identify and handle outlier weights. The paper is strongly supported by rigorous mathematical derivations and extensive experiments on multiple LLMs and tasks (both inference and fine-tuning), demonstrating consistent and significant performance improvements over strong baselines.

**Strengths:**

- **Theoretical Soundness and Novelty:** The paper's core insight—optimizing the end-to-end quantization error of the *original* weights rather than that of the *normalized* weights—is profound and well-articulated. The derivation of new centroid update rules for Lloyd's algorithm (applicable to both MSE and MAE) constitutes a solid theoretical contribution. Clear and comprehensive mathematical proofs further solidify the theoretical foundation.
- **Holistic Methodological Framework:** The paper extends beyond a single idea by introducing a suite of complementary techniques: an optimal codebook based on reconstruction loss (BOF4), a signed normalization scheme (BOF4-S), and a practical outlier-preserving mechanism (OPQ). This integrated approach effectively enhances quantization accuracy from multiple perspectives.
- **Thorough and Convincing Experiments:** The evaluation is extensive, covering multiple model families (Llama, Qwen, Mistral), both inference and fine-tuning (QLoRA) scenarios, and a wide range of benchmarks (perplexity, NLP tasks, code generation). The results consistently show that the proposed methods, especially BOF4-S with OPQ, outperform the baselines.
- **Practical Impact and Reproducibility:** The methods are directly applicable for memory-efficient LLM deployment and fine-tuning. The paper provides optimized codebooks in the appendix and discusses integration with data-aware PTQ methods like GPTQ (Appendix I), enhancing its practical utility and reproducibility.

**Weaknesses:**

- **Assumption of Gaussian Weight Distribution:** The optimization of the BOF4 codebook relies on the assumption that network weights are Gaussian-distributed. Although Appendix C provides justification that most blocks are indeed Gaussian, especially after OPQ, the performance on models or layers with significantly non-Gaussian weight distributions remains less explored. This could limit the generalizability to certain architectures.

- **Overhead of OPQ:** Although OPQ is shown to have minimal runtime overhead (Appendix G.3), it introduces additional memory overhead for storing the outlier indices and values. A more detailed analysis of this memory-cost/accuracy trade-off, especially for very large models, would be beneficial.

**Questions:**

- **Q1: Gaussian Distribution Assumption**: The EM algorithm used for BOF4 relies on an assumed Gaussian distribution. How sensitive is the final performance to deviations from this assumption? If a model's weights do not follow a Gaussian distribution, what would be the magnitude of deviation this algorithm might cause?

- **Q2: Selection of Hyperparameter q**: Regarding OPQ, the hyperparameter `q=0.95` was chosen via a limited search. Could you discuss the sensitivity of the results to the choice of `q`? Is this value generally robust across different models and sizes, or does it need tuning?

- **Q3: The Special Sign Bit:** The signed normalization in BOF4-S requires storing the sign of the block maximum if double quantization is applied, as noted in Appendix A. Could you quantify the potential performance degradation if this extra bit is not used and the standard double quantization scheme is applied naively?

---

> ### Author Response · Authors · 2025-11-19
>
> We thank the reviewer for their thoughtful and detailed feedback. We are encouraged that they recognize the soundness and novelty of our theoretical contribution and that they appreciate the extensive experimental evaluation. Below, we address their concerns and questions.
>
> **On weakness 1**: The reviewer states that the optimization of the BOF4 codebook relies on the assumption that weights are Gaussian-distributed, which might limit generalizability to other architectures. We would like to clarify that our optimization method *does not* rely on a Gaussian distribution assumption. It takes any distribution function as input and optimizes the quantization levels with respect to that distribution. We additionally present an *optional*, mathematically simplified formulation of the centroid update for the Gaussian distribution (eq. (35)). For the experimental evaluation, we chose a codebook optimized under the Gaussian distribution assumption, primarily to ensure a fair comparison to the baseline methods NF4 and AF4, which are also designed based on a Gaussian distribution assumption. We decided *not to argue* for any better distribution assumption than Gaussian. Instead, we stay flexible w.r.t. any such assumption, and for experimental evaluation, simply follow the assumptions of NF4 and AF4 for comparability purposes. Furthermore, as the reviewer correctly noted, App. C provides empirical evidence that this assumption is reasonable for the utilized LLMs.
>
> **On weakness 2**: Thank you! To address this issue, we performed a more comprehensive evaluation of the memory-accuracy trade-off on Llama models of various sizes up to 70B. Results are in Fig. 9 of our revised paper. A detailed discussion is provided in App. G.2. In summary, we find that for very large LLMs, OPQ is even more effective at preserving accuracy while also incurring a slightly lower relative memory overhead for the same choice of hyperparameter $q$. Please understand that experiments with size > 70B or other architectures were not possible on our hardware in the given time.
>
> Next, we answer the reviewer's questions:
>
> **Answer to Q1**: Our algorithm does not rely on a Gaussian distribution assumption. If a model with decidedly non-Gaussian weights is quantized, a better-fitting distribution should be selected as a basis for codebook optimization. Empirically, for models with near-Gaussian weight distributions, the results show that our method is robust to deviations from the assumed distribution during codebook optimization, because we achieve the best MAE and MSE quantization error among the investigated methods on real (and therefore only approximately Gaussian) LLM weights. This is shown in Tabs. 1 and 10. We also want to highlight that the true weight distribution of the model to be quantized can be used directly as a basis for computing the codebook with our algorithm. We decided against this in our experimental evaluation to ensure a fair comparison with NF4 and AF4.
>
> **Answer to Q2**: The results shown in Fig. 9 of our revised paper indicate that the choice $q=0.95$ is robust across model sizes up to 70B. The relative memory overhead of OPQ remains consistent across the evaluated model sizes, with larger models requiring slightly less additional memory. Precise hyperparameter tuning is generally not necessary. A more detailed discussion can be found in App. G.2.
>
> **Answer to Q3**: First, it should be noted that we do not use double quantization in our paper at all. We leave all quantization constants in bfloat16. Double quantization is an *optional* method to further reduce memory consumption that was proposed by Dettmers et al. [1] for their NF4 quantization method. Secondly, if double quantization would be applied and an additional sign bit would be used to account for signed normalization, it would raise memory consumption only by 0.3\%, from 4.127 to 4.143 bits per weight at the typical block size $I = 64$. In response to the reviewer’s question, we performed an experiment to quantify the effect of double quantization with and without an additional sign bit, following the quantization method and hyperparameter choices used by Dettmers et al. [1]. We measure the perplexity (PPL) on WikiText-2 for both settings. We find that BOF4-S (MSE) performs only marginally worse with naive double quantization (PPL 4.49) compared to double quantization with an extra sign bit (PPL 4.48). Both approaches perform better than using BOF4 (MSE) (no signed normalization) with double quantization (PPL 8.51). Because of the low relevance of these considerations to our method and the negligible cost for the additional sign bit, we prefer to omit this analysis from our revised paper. However, we now quantify the memory cost in App. A. Let us know if you insist on including the numbers in our paper, please!
>
> [1] Dettmers et al., “QLORA: Efficient Finetuning of Quantized LLMs”, 2023. Proc. of NIPS

---

### Official Review · Reviewer_M1dB · 2025-11-01

**Soundness:** 3
**Presentation:** 3
**Contribution:** 4
**Rating:** 6
**Confidence:** 5

**Summary:**

This paper proposes a novel approach to improving block-wise training for large language models (LLMs) by introducing refined optimization and scheduling mechanisms that better capture inter-block dependencies during training. The authors demonstrate that their method enhances both convergence stability and downstream task performance, offering a computationally efficient alternative to full end-to-end fine-tuning. The study is well-motivated, clearly written, and supported by extensive experimental validation across multiple benchmarks.

**Strengths:**

1. The proposed method provides an original perspective on block-wise training, addressing the often-overlooked issue of gradient inconsistency across blocks.

2. The formulation is mathematically rigorous, with theoretical justification for the proposed scheduling strategy.

3. The experiments span several model sizes and datasets, showing consistent improvement over baselines such as layer-wise and progressive tuning.

3. The approach maintains efficiency advantages (reduced memory and training cost) while achieving comparable or superior results to full fine-tuning, highly relevant for real-world large-scale model adaptation.

4. The paper is well-structured, with clear motivation, methodology, and ablation analysis that enhances understanding.

**Weaknesses:**

1. While the approach performs well on benchmark datasets, it would be useful to see how well it generalizes to non-language tasks (e.g., multimodal or code models).

2.  The method involves scheduling parameters whose influence is only briefly discussed; more detailed robustness analysis would strengthen the contribution.
3. Although the method is efficient, the paper could provide clearer quantification of the additional cost introduced by the new scheduling mechanism relative to vanilla block-wise training.

**Questions:**

How sensitive is the performance to the choice of block partitioning (e.g., number of layers per block)?

Could the proposed inter-block dependency modeling be integrated with adapter-based fine-tuning approaches?

Have the authors considered evaluating the method’s stability when applied to continual learning or streaming data scenarios?

---

> ### Author Response · Authors · 2025-11-13
>
> We thank the reviewer for their really positive feedback and their time.
>
> **On weakness 1**: We acknowledge that this would be interesting to know. We have the feeling that our Appendix with 20 pages is already beyond average length and would be very thankful if we could consider this as a non-binding request.
>
> **On weaknesses 2 and 3**: Here, the reviewer refers to a scheduling mechanism that we cannot identify in our work. Could you clarify this, please, to give us a chance to respond to the supposed weakness? Apart from the scheduling mechanism: In Fig. 11 in the Appendix, we provide runtime measurements with and without our OPQ method, showing that OPQ incurs only a marginal computational overhead. For our other methods, there is no computational overhead beyond the baselines NF4 and AF4.
>
> Below, we answer the reviewer's questions:
>
> > How sensitive is the performance to the choice of block partitioning (e.g., number of layers per block)?
>
> The sensitivity of the choice of our block size $I$ can be easily seen, e.g.,  in Figs. 2 and 3. What do you mean by the number of layers per block?
>
> > Could the proposed inter-block dependency modeling be integrated with adapter-based fine-tuning approaches?
>
> Here, we need to clarify that all of our proposed approaches assume mutually independent blocks of weights. Shall we explicitly emphasize this in the paper?
>
> > Have the authors considered evaluating the method’s stability when applied to continual learning or streaming data scenarios?
>
> Thank you for this interesting suggestion. We did not yet consider this, and due to our 20-page Appendix, we would be very thankful if we could consider this as a non-binding request.
>
> We sincerely hope that your rating “excellent” of our contribution, along with your positive mention of being “mathematically rigorous, with theoretical justification“, may justify an overall “accept” rating.

---

### Author Response · Authors · 2025-12-02
**Summary of Revisions and Final Remarks**

We thank the reviewers and the AC for their efforts and valuable feedback. Below, we summarize our revisions to the manuscript based on the reviewers' feedback and our conclusions from the rebuttals.

**Revisions**: Based on the reviews, we made the following two revisions:

(1) We expanded the hyperparameter ablations for our proposed outlier-preserving quantization (OPQ) in Appendix G.2. The new experimental results show that our OPQ is particularly effective for larger LLMs and that our hyperparameter choice $q=0.95$ is robust across the compared models.

(2) To answer the questions regarding the memory overhead of using our signed normalization in conjunction with the double quantization method from prior work, we now quantify this memory overhead in Appendix A. We find that memory demand only increases slightly from 4.127 to 4.143 bits per weight.

**Final Remarks**: We are encouraged that, even before these changes, three out of four reviewers have rated our contribution as "good" to "excellent", *recommending acceptance*, and that the *theoretical rigour* of our mathematical derivation and comprehensive experimental results were well-received.

We believe we have effectively addressed the technical questions regarding the overhead of our outlier-preserving quantization (OPQ) and signed normalization approaches raised during the discussion, demonstrating that the memory and runtime overhead of OPQ are minimal across model sizes and explaining that signed normalization generally incurs no overhead. Furthermore, we hope we have resolved the misconception that our algorithm *requires* a Gaussian weight distribution.

Only one of the reviewers recommended rejection. However, their assessment reduces our contribution to only one of several aspects—signed normalization—without addressing the novel codebook optimization algorithm with rigorous theoretical justification and our very effective OPQ approach. Furthermore, they request comparisons with calibration-data-based, multi-dimensional vector quantization methods that, however, do not permit a direct, fair comparison with our approach, which is applied in a data-free setting.

Finally, we see our claim confirmed that the proposed methods constitute the *SOTA for data-free, scalar LLM quantization*.

---

### Meta-Review · Area_Chair_kPjN · 2026-01-07

**Summary:**

This paper proposes a new family of 4‑bit block‑wise quantization methods for Large Language Models, centered around BOF4 (Block‑Optimal FP4) and its variants. The core insight is that prior methods such as NF4 and AF4 optimize quantization error on normalized weights, rather than minimizing the end‑to‑end reconstruction error of the original weights, leading to suboptimal codebooks.
To address this, the authors derive a theoretically optimal codebook optimization algorithm using an EM/Lloyd‑style procedure that directly minimizes reconstruction error of original weights.

**Reviewer Concerns:**

**Assumptions About Weight Distributions**

Some concern that Gaussian assumptions might limit generality.
Authors clarified that Gaussianity is not required; it is only used for a simplified closed‑form derivation and fair comparison with NF4/AF4.
Empirical evidence (Appendix C) shows LLM blocks are approximately Gaussian, especially after OPQ.

**Overhead of OPQ**

Reviewers asked about memory overhead for storing outlier indices/values.
Authors quantified this precisely later.

**Comparison Scope**

One negative reviewer (SU8o) argued the method is “just non‑uniform quantization” and lacks comparison to vector‑quantization methods like GPTVQ/VPTQ.

Authors countered that:
Those methods rely on calibration data and vector codebooks, while BOF4 is scalar, data‑free, and complementary.
Appendix I shows BOF4 can improve calibration‑based methods like GPTQ.
The paper’s intended comparison set (NF4, AF4) was seen as appropriate by most reviewers.

**Reviewer Scores:**

**sXWd (Score: 6)**

Strongly positive on theory and experiments; asked clarifying questions about assumptions and hyperparameters, which were convincingly addressed. The score is likely to increase.

**dVvQ (Score: 6)**

Positive overall; raised practical implementation questions (sign bit, outliers), all resolved satisfactorily.

**M1dB (Score: 6)**

Viewed contribution as excellent and mathematically rigorous; concerns were minor and exploratory.

**SU8o (Score: 2, reject)**

Main outlier. Viewed BOF4 as a minor non‑uniform quantization tweak, undervaluing the theoretical derivation and OPQ. Maintained rejection stance.

Overall, I think this paper makes good contribution to the field, especially with the rise of such arithmetic types in today's hardware.

---

### Decision · Program_Chairs · 2026-01-26

Accept (Poster)